# SI$\mathcal{X}$O: Smoothing Inference with Twisted Objectives

**Dieterich Lawson**,[*] **Allan Raventós**,[*] **Andrew Warrington**,[*] **Scott Linderman**

{jdlawson, aravento, awarring, scott.linderman}@stanford.edu

Stanford University

## Abstract

Sequential Monte Carlo (SMC) is an inference algorithm for state space models that approximates the posterior by sampling from a sequence of target distributions. The target distributions are often chosen to be the *filtering* distributions, but these ignore information from future observations, leading to practical and theoretical limitations in inference and model learning. We introduce *SIXO*, a method that instead learns target distributions that approximate the *smoothing* distributions, incorporating information from all observations. The key idea is to use density ratio estimation to fit functions that warp the filtering distributions into the smoothing distributions. We then use SMC with these learned targets to define a variational objective for model and proposal learning. SIXO yields provably tighter log marginal lower bounds and offers more accurate posterior inferences and parameter estimates in a variety of domains.

## 1 Introduction

In this work we consider model learning and approximate posterior inference in probabilistic state space models. Sequential Monte Carlo (SMC) is a general-purpose method for these problems [1–3] that produces an unbiased estimate of the marginal likelihood as well as latent state trajectories (i.e. *particles*) that can be used to approximate posterior expectations. SMC can facilitate model learning via expectation-maximization or direct maximization of the marginal likelihood estimate [4, 5]. It can also be cast in a variational framework [6, 7] as a rich family of approximate posterior distributions that can be fit using stochastic gradient ascent and modern automatic differentiation methods [8–12].

The quality of SMC's marginal likelihood and posterior estimates is driven by two design decisions: the choice of *proposal distributions* and *target distributions*. The proposal distributions specify how particles propagate from one time step to the next, while the target distributions specify how those particles are weighted and which ones survive to future time steps. The most common SMC variant, filtering SMC, sets the targets to the *filtering distributions*, the conditional distributions over latent states $\mathbf{x}_{1:t} = (\mathbf{x}_1, \ldots, \mathbf{x}_t)$ given observations $\mathbf{y}_{1:t} = (\mathbf{y}_1, \ldots, \mathbf{y}_t)$. The central issue is that the filtering distributions do not incorporate information from future observations $\mathbf{y}_{t+1:T}$.

Figure 1 illustrates why setting the target distributions to the filtering distributions can be problematic. In this example, the latent states follow a simple Gaussian random walk, but only the last step is observed. Thus, the filtering distributions reduce to the prior, a series of mean zero Gaussians, shown in Figure 1a. If the observation is far from the prior, the filtering distribution suddenly jumps at time $T = 10$. This is a recipe for disaster in SMC: the particles at time $T - 1$ will be distributed according to a mean zero Gaussian and very few will survive to the next time step, causing the variance of the SMC estimator to explode. Even if the proposals incorporate smoothing information, using filtering targets can cause particle degeneracy by resampling away high-quality particles, as seen in Figure 1c.

---

[*]Equal contribution.

36th Conference on Neural Information Processing Systems (NeurIPS 2022).

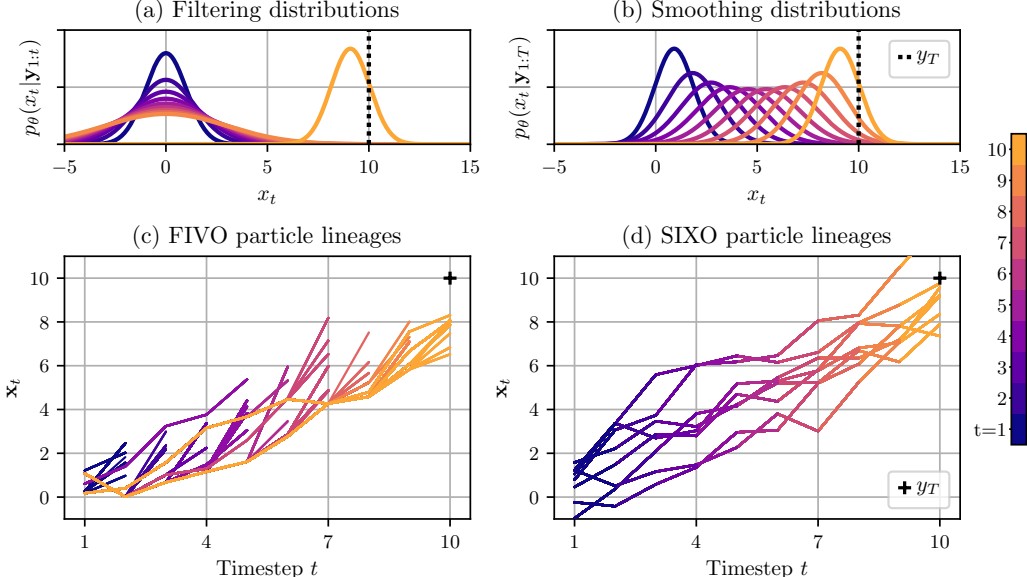

**Figure 1:** Theoretical and empirical target distributions for a Gaussian random walk with a single observation $y_T = 10$ and $\mathbf{y}_{1:T-1} = \varnothing$. **(a)** For $t = 1, \dots, 9$ the filtering distributions reduce to a series of mean-zero Gaussians. At $t = T = 10$, the filtering distribution incorporates the observation $y_T$, resulting in a sudden shift and particle death. **(b)** In contrast, the smoothing distributions steadily shift towards the observation, matching the posterior perfectly. **(c)** The proposal learned by a previous method, FIVO [8–10], exploits smoothing information to propose particles upwards towards the observed value. However, FIVO is based on filtering SMC which "resists" this by resampling particles back towards the prior, resulting in particle degeneracy. **(d)** SIXO's proposal also leverages smoothing information, but proposed particles are preserved by the learned target distributions.

Suppose instead that the target distributions were the *smoothing distributions*—the conditional distributions over latents $\mathbf{x}_{1:t}$ given *all* observations $\mathbf{y}_{1:T}$. Figure 1b shows the smoothing distributions for the simple Gaussian random walk. Unlike the filtering distributions, the smoothing distributions shift steadily toward the observation over time. These slow, smooth changes are ideal for SMC: Figure 1d shows many particles surviving from one step to the next, resulting in a low-variance SMC estimator.

In practice we do not have access to the smoothing distributions—if we did, there would be no need for SMC! Here, we introduce a new method called SIXO: Smoothing Inference with Twisted Objectives. SIXO provides a unified approach for learning model and proposal parameters, as well as a set of *twisting functions* that warp the filtering distributions into targets that better approximate the smoothing distributions [13]. Like its predecessor FIVO [8–10], SIXO uses a variational approach, deriving a lower bound to the marginal likelihood. Unlike its predecessor, we prove that the SIXO bound can become tight, even with finitely many particles.

The key challenge with SIXO is learning the twisting functions. We find that a simple density ratio estimation approach works best, and we propose an algorithm that interleaves twist updates with updates to the model and proposal. Thus, SIXO offers a means of jointly learning model parameters, SMC proposals, and targets for accurate posterior inference.

Finally, we give empirical evidence to support our theoretical claims. Across a range of experiments with a Gaussian diffusion, a stochastic volatility model of currency exchange rates, and a Hodgkin-Huxley model of membrane potential in a neuron, SIXO consistently outperforms FIVO and related methods. We dissect these results to illustrate how learning better targets enables more effective posterior inference and model learning.

## 2 Background

Consider modeling sequential data $\mathbf{y}_{1:T} \in \mathcal{Y}^T$ using latent variables $\mathbf{x}_{1:T} \in \mathcal{X}^T$ with Markovian structure, and let the joint distribution factorize as

$$p_{\boldsymbol{\theta}}(\mathbf{x}_{1:T}, \mathbf{y}_{1:T}) = p_{\boldsymbol{\theta}}(\mathbf{x}_1) p_{\boldsymbol{\theta}}(\mathbf{y}_1 \mid \mathbf{x}_1) \prod_{t=2}^{T} p_{\boldsymbol{\theta}}(\mathbf{x}_t \mid \mathbf{x}_{t-1}) p_{\boldsymbol{\theta}}(\mathbf{y}_t \mid \mathbf{x}_t) \tag{1}$$

with global parameters $\boldsymbol{\theta} \in \Theta$. We further assume that the conditional distributions $p_{\boldsymbol{\theta}}(\mathbf{x}_t \mid \mathbf{x}_{t-1})$ and $p_{\boldsymbol{\theta}}(\mathbf{y}_t \mid \mathbf{x}_t)$ may depend nonlinearly on $\mathbf{x}_{t-1}$ and $\mathbf{x}_t$ respectively.

The marginal likelihood and posterior for this model class are not readily available from the joint distribution due to the intractable integral over the latents $\mathbf{x}_{1:T}$, i.e.

$$p_{\boldsymbol{\theta}}(\mathbf{y}_{1:T}) = \int_{\mathcal{X}^T} p_{\boldsymbol{\theta}}(\mathbf{y}_{1:T}, \mathbf{x}_{1:T}) \, \mathrm{d}\mathbf{x}_{1:T},$$

cannot easily be computed due to the form of the conditional distributions.

### 2.1 Sequential Monte Carlo

Sequential Monte Carlo is an algorithm for inference in state-space models that approximates the posterior $p_{\boldsymbol{\theta}}(\mathbf{x}_{1:T} \mid \mathbf{y}_{1:T})$ with a set of $K$ weighted particles $\mathbf{x}_{1:T}^{1:K}$. These particles are constructed by approximately sampling from a sequence of target distributions $\{\pi_t(\mathbf{x}_{1:t})\}_{t=1}^{T}$, with the intuition that sampling from a series of distributions that gradually approach the posterior is easier than attempting to sample from it directly. The targets are often only available up to an unknown normalizing constant $Z_t$, so SMC uses the unnormalized targets $\{\gamma_t(\mathbf{x}_{1:t})\}_{t=1}^{T}$, which correspond to the normalized targets via $\pi_t(\mathbf{x}_{1:t}) = \gamma_t(\mathbf{x}_{1:t})/Z_t$.

SMC repeats three steps: First, a set of latents are sampled from a proposal distribution $q_{\boldsymbol{\theta}}(\mathbf{x}_t^k \mid \mathbf{x}_{t-1}^k, \mathbf{y}_{1:T})$ conditional on the current particles $\mathbf{x}_{1:t-1}^{1:K}$. Then, each particle is weighted using the unnormalized target $\gamma_t(\mathbf{x}_{1:t})$ to form an empirical approximation of the normalized target distribution. Finally, new particle trajectories $\mathbf{x}_{1:t}^{1:K}$ are drawn from this approximation to the normalized target.

Ideally the target distributions smoothly approach the posterior so that sampling from the target at time $t+1$ is easy given samples from the target at time $t$. As long as mild technical conditions are met and $\gamma_T(\mathbf{x}_{1:T}) \propto p_{\boldsymbol{\theta}}(\mathbf{x}_{1:T}, \mathbf{y}_{1:T})$, SMC returns a consistent and unbiased estimate of the marginal likelihood $p_{\boldsymbol{\theta}}(\mathbf{y}_{1:T})$ and a set of weighted particles approximating the posterior $p_{\boldsymbol{\theta}}(\mathbf{x}_{1:T} \mid \mathbf{y}_{1:T})$ [1–3]. For more details see Appendix B.1, and for a thorough treatment of SMC see [1–3].

### 2.2 Filtering SMC and Model Learning

The most commonly-used SMC algorithm is filtering SMC (FSMC), which sets the normalized targets to the *filtering* distributions, i.e. $\pi_t(\mathbf{x}_{1:t}) = p_{\boldsymbol{\theta}}(\mathbf{x}_{1:t} \mid \mathbf{y}_{1:t})$ and $\gamma_t(\mathbf{x}_{1:t}) \propto p_{\boldsymbol{\theta}}(\mathbf{x}_{1:t}, \mathbf{y}_{1:t})$. Let $\widehat{Z}_{\text{FSMC}}(\boldsymbol{\theta}, \mathbf{y}_{1:T})$ be the marginal likelihood estimator returned from running filtering SMC with proposal distributions $\{q_{\boldsymbol{\theta}}(\mathbf{x}_t \mid \mathbf{x}_{1:t-1}, \mathbf{y}_{1:t})\}_{t=1}^{T}$ which may share parameters with $p_{\boldsymbol{\theta}}$.

Previous work used filtering SMC to fit model parameters by ascending a lower bound on the log marginal likelihood called a *filtering variational objective* (FIVO) [8–10]. The FIVO bound is derived using Jensen's inequality and the unbiasedness of $\widehat{Z}_{\text{FSMC}}$,

$$\mathcal{L}_{\text{FIVO}}(\boldsymbol{\theta}, \mathbf{y}_{1:T}) \triangleq \mathbb{E}[\log \widehat{Z}_{\text{FSMC}}(\boldsymbol{\theta}, \mathbf{y}_{1:T})] \leq \log \mathbb{E}[\widehat{Z}_{\text{FSMC}}(\boldsymbol{\theta}, \mathbf{y}_{1:T})] = \log p_{\boldsymbol{\theta}}(\mathbf{y}_{1:T}), \tag{2}$$

and is optimized using stochastic gradient ascent in $\boldsymbol{\theta}$ [8–10, 14].

### 2.3 Smoothing SMC via Twisting Functions

The main disadvantage of filtering SMC is that the filtering distributions only condition on observations up to the current timestep $t$, ignoring future observations $\mathbf{y}_{t+1:T}$. This creates situations where future observations are highly unlikely given the current latent trajectories, which in turn causes particle death, high variance normalizing constant estimates, and poor inference and model learning [8, 13, 15]. Performing smoothing SMC would resolve this issue by choosing the *smoothing*

distributions as targets, i.e. $\pi_t(\mathbf{x}_{1:t}) = p_{\boldsymbol{\theta}}(\mathbf{x}_{1:t} \mid \mathbf{y}_{1:T})$ and $\gamma_t(\mathbf{x}_{1:t}) \propto p_{\boldsymbol{\theta}}(\mathbf{x}_{1:t}, \mathbf{y}_{1:T})$. Unfortunately, $p_{\boldsymbol{\theta}}(\mathbf{x}_{1:t}, \mathbf{y}_{1:T})$ is not readily available from the model and computing it is roughly as hard as the original inference problem.

However, $p_{\boldsymbol{\theta}}(\mathbf{x}_{1:t}, \mathbf{y}_{1:T})$ factors into the product of the filtering distributions, $p_{\boldsymbol{\theta}}(\mathbf{x}_{1:t}, \mathbf{y}_{1:t})$, and the *lookahead* distributions, $p_{\boldsymbol{\theta}}(\mathbf{y}_{t+1:T} \mid \mathbf{x}_t)$ (Appendix B.2). If the lookahead distributions can be well-approximated by a series of "twisting" functions [13], $\{r(\mathbf{y}_{t+1:T}, \mathbf{x}_t)\}_{t=1}^T$, then running SMC with targets $\gamma_t(\mathbf{x}_{1:t}) = p_{\boldsymbol{\theta}}(\mathbf{x}_{1:t}, \mathbf{y}_{1:t})r(\mathbf{y}_{t+1:T}, \mathbf{x}_t)$ would approximate smoothing SMC. In this sense, the lookahead distributions are optimal twisting functions [13, 16].

Different twisting functions yield different SMC methods such as auxiliary particle filters and twisted particle filters [13, 17]. However, as long as the final unnormalized target $\gamma_T(\mathbf{x}_{1:T})$ is proportional to $p_{\boldsymbol{\theta}}(\mathbf{x}_{1:T}, \mathbf{y}_{1:T})$ and regularity conditions are met, SMC will produce an unbiased estimate of the marginal likelihood, regardless of the choice of twisting functions [1, 3, 18]. Instead, the quality of the twisting functions affects the variance of SMC's marginal likelihood estimate.

# 3 SIXO: Smoothing Inference with Twisted Objectives

Our goal is to fit models by optimizing a lower bound on their log marginal likelihood constructed using smoothing SMC. To construct the lower bound, fix $r_{\boldsymbol{\psi}}(\mathbf{x}_T) = 1$ and let $\widehat{Z}_{\text{SIXO}}(\boldsymbol{\theta}, \boldsymbol{\psi}, \mathbf{y}_{1:T})$ be the marginal likelihood estimator returned from running SMC with unnormalized targets $\{p_{\boldsymbol{\theta}}(\mathbf{x}_{1:t}, \mathbf{y}_{1:t})r_{\boldsymbol{\psi}}(\mathbf{y}_{t+1:T}, \mathbf{x}_t)\}_{t=1}^T$ and proposal distributions $\{q_{\boldsymbol{\theta}}(\mathbf{x}_t \mid \mathbf{x}_{1:t-1}, \mathbf{y}_{1:T})\}_{t=1}^T$. Because the $T^{\text{th}}$ unnormalized target is $p_{\boldsymbol{\theta}}(\mathbf{x}_{1:T}, \mathbf{y}_{1:T})$, $\widehat{Z}_{\text{SIXO}}$ will be an unbiased estimator of the marginal likelihood $p_{\boldsymbol{\theta}}(\mathbf{y}_{1:T})$ [1, 3]. This implies via Jensen's inequality that

$$\begin{aligned}
\mathcal{L}_{\text{SIXO}}(\boldsymbol{\theta}, \boldsymbol{\psi}, \mathbf{y}_{1:T}) &\triangleq \mathbb{E}\left[\log \widehat{Z}_{\text{SIXO}}(\boldsymbol{\theta}, \boldsymbol{\psi}, \mathbf{y}_{1:T})\right] \\
&\leq \log \mathbb{E}\left[\widehat{Z}_{\text{SIXO}}(\boldsymbol{\theta}, \boldsymbol{\psi}, \mathbf{y}_{1:T})\right] = \log p_{\boldsymbol{\theta}}(\mathbf{y}_{1:T})
\end{aligned} \tag{3}$$

i.e. $\mathcal{L}_{\text{SIXO}}(\boldsymbol{\theta}, \boldsymbol{\psi}, \mathbf{y}_{1:T})$ is a lower bound on the log marginal likelihood $\log p_{\boldsymbol{\theta}}(\mathbf{y}_{1:T})$ [14].

## 3.1 The Functional Form of the Twists

The structure of the lookahead distributions $p_{\boldsymbol{\theta}}(\mathbf{y}_{t+1:T} \mid \mathbf{x}_t)$ suggests a functional form for $r_{\boldsymbol{\psi}}$ that accepts a single latent $\mathbf{x}_t$ and produces distributions over all future observations $\mathbf{y}_{t+1:T}$. Because the twists will be evaluated once per particle and timestep in an SMC sweep, this functional form would lead to an algorithm with $O(T^2)$ complexity. To reduce the complexity, we consider two methods: fixed-lag twisting and backwards twisting.

**Fixed-lag twisting** approximates the full lookahead distribution $p_{\boldsymbol{\theta}}(\mathbf{y}_{t+1:T} \mid \mathbf{x}_t)$ using a fixed window of $L$ observations, i.e. it models $p_{\boldsymbol{\theta}}(\mathbf{y}_{t+1:t+L} \mid \mathbf{x}_t)$ [17, 19, 20]. We define the fixed-lag twisting functions $\{r_{\boldsymbol{\psi}}(\mathbf{y}_{t+1:t+L}, \mathbf{x}_t)\}_{t=1}^{T-1}$ as a sequence of functions which accept $\mathbf{x}_t \in \mathcal{X}$ and produce a distribution over $\mathbf{y}_{t+1:t+L} \in \mathcal{Y}^L$. This reduces the computational complexity to $O(TL)$, at the expense of only looking at $L$ observations.

In our experiments we use an $L = 1$ twist that scores the next observation by approximating the one-step lookahead

$$p_{\boldsymbol{\theta}}(\mathbf{y}_{t+1} \mid \mathbf{x}_t) = \int p_{\boldsymbol{\theta}}(\mathbf{y}_{t+1} \mid \mathbf{x}_{t+1}) \, p_{\boldsymbol{\theta}}(\mathbf{x}_{t+1} \mid \mathbf{x}_t) \, \mathrm{d}\mathbf{x}_{t+1} \tag{4}$$

with Gauss-Hermite quadrature [21]. We refer to this as the "quadrature twist". This approach is similar to the APF [17, 22], but uses numerical quadrature in place of sample-based integration.

**Backwards twisting** is motivated by rewriting the lookahead distributions using Bayes' rule,

$$p_{\boldsymbol{\theta}}(\mathbf{y}_{t+1:T} \mid \mathbf{x}_t) = \frac{p_{\boldsymbol{\theta}}(\mathbf{x}_t \mid \mathbf{y}_{t+1:T}) \, p_{\boldsymbol{\theta}}(\mathbf{y}_{t+1:T})}{p_{\boldsymbol{\theta}}(\mathbf{x}_t)} \propto \frac{p_{\boldsymbol{\theta}}(\mathbf{x}_t \mid \mathbf{y}_{t+1:T})}{p_{\boldsymbol{\theta}}(\mathbf{x}_t)}, \tag{5}$$

where we drop terms independent of $\mathbf{x}_t$ because the twisting functions will be used to score particles in SMC. Thus, we need only approximate $p_{\boldsymbol{\theta}}(\mathbf{x}_t \mid \mathbf{y}_{t+1:T})/p_{\boldsymbol{\theta}}(\mathbf{x}_t)$. The numerator $p_{\boldsymbol{\theta}}(\mathbf{x}_t \mid \mathbf{y}_{t+1:T})$

---

**Algorithm 1** SIXO-DRE

1: **procedure** SIXO-DRE($\mathbf{y}_{1:T}, \boldsymbol{\theta}_0, \boldsymbol{\psi}_0, S, N, K$)
2:    **for** $s = 1, \ldots, S$ **do**
3:       $\boldsymbol{\psi}_s = $TWIST-UPDATE($\boldsymbol{\theta}_{s-1}, \boldsymbol{\psi}_{s-1}, N$)
4:       $\boldsymbol{\theta}_s = $MODEL-UPDATE($\mathbf{y}_{1:T}, \boldsymbol{\theta}_{s-1}, \boldsymbol{\psi}_s, N, K$)
5:    **return** $\boldsymbol{\theta}_S, \boldsymbol{\psi}_S$

6: **procedure** TWIST-UPDATE($\boldsymbol{\theta}, \boldsymbol{\psi}_0, N$)
7:    **for** $i = 1, \ldots, N$ **do**
8:       $\tilde{\mathbf{x}}_{1:T} \sim p_{\boldsymbol{\theta}}(\mathbf{x}_{1:T})$
9:       $\mathbf{x}_{1:T}, \mathbf{y}_{1:T} \sim p_{\boldsymbol{\theta}}(\mathbf{x}_{1:T}, \mathbf{y}_{1:T})$
10:      $\mathcal{L}_{\text{DRE}}(\boldsymbol{\psi}) = \frac{1}{T-1} \sum_{t=1}^{T-1} \log \sigma(\log r_{\boldsymbol{\psi}}(\mathbf{y}_{t+1:T}, \mathbf{x}_t)) + \log(1 - \sigma(\log r_{\boldsymbol{\psi}}(\mathbf{y}_{t+1:T}, \tilde{\mathbf{x}}_t)))$
11:      Compute $\boldsymbol{\psi}_i$ using the gradients of $\mathcal{L}_{\text{DRE}}$ evaluated at $\boldsymbol{\psi}_{i-1}$
12:    **return** $\boldsymbol{\psi}_N$

13: **procedure** MODEL-UPDATE($\mathbf{y}_{1:T}, \boldsymbol{\theta}_0, \boldsymbol{\psi}, N, K$)
14:    **for** $i = 1, \ldots, N$ **do**
15:      $\widehat{Z}_{\text{SIXO}}(\boldsymbol{\theta}) = \text{SMC}(\{p_{\boldsymbol{\theta}}(\mathbf{x}_{1:t}, \mathbf{y}_{1:t}) r_{\boldsymbol{\psi}}(\mathbf{y}_{t+1:T}, \mathbf{x}_t)\}_{t=1}^T, \{q_{\boldsymbol{\theta}}(\mathbf{x}_t \mid \mathbf{x}_{t-1}, \mathbf{y}_{1:T})\}_{t=1}^T, \text{K})$
16:      Compute $\boldsymbol{\theta}_i$ using the biased gradients of $\widehat{Z}_{\text{SIXO}}$ evaluated at $\boldsymbol{\theta}_{i-1}$
17:    **return** $\boldsymbol{\theta}_N$

18: **procedure** SMC($\{\gamma_t(\mathbf{x}_{1:t})\}_{t=1}^T, \{q_{\boldsymbol{\theta}}(\mathbf{x}_t \mid \mathbf{x}_{t-1}, \mathbf{y}_{1:T})\}_{t=1}^T, \text{K}$)
19:    See Algorithm 2 in Appendix B.1.

---

is the reverse of the lookahead distributions—it is a distribution over a single latent conditioned on future observations. This makes it possible to parameterize the twists using a recurrent function approximator (e.g. a recurrent neural network or RNN) run backwards across the observations $\mathbf{y}_{1:T}$ to produce twist values for each timestep as a function of $\mathbf{x}_t^k$.

We define the backwards twists $\{r_{\boldsymbol{\psi}}(\mathbf{y}_{t+1:T}, \mathbf{x}_t)\}_{t=1}^{T-1}$ as a sequence of positive, integrable, real-valued functions $\mathcal{Y}^{T-t} \times \mathcal{X} \to \mathbb{R}_+$ with parameters $\boldsymbol{\psi} \in \boldsymbol{\Psi}$. Parameterizing backward twists with a recurrent function approximator results in $O(T)$ complexity (see Appendix C.3) and allows the twist to condition on all future observations, making backwards twisting preferable to fixed-lag twisting.

## 3.2 Learning Twists

**Ascending the Unified Objective**    One way to fit the twists, proposal, and model is to ascend $\mathcal{L}_{\text{SIXO}}$ in the parameters of $p_{\boldsymbol{\theta}}, q_{\boldsymbol{\theta}}$, and $r_{\boldsymbol{\psi}}$, similar to FIVO [8–10]. The gradients of this objective include score-function terms that arise from the discrete resampling steps in SMC. We refer to ascending $\mathcal{L}_{\text{SIXO}}$ with these unbiased gradients as SIXO-u. Because the resampling gradient terms have high variance, SIXO-u is impractical for complex settings [8, 11]. Lower-variance methods for estimating these gradients were explored by Lawson et al. [11] but found to be ineffective. We therefore seek an alternative method for training the twists. For a detailed discussion and derivation of the gradient, see Appendix C.1.

**Density Ratio Estimation**    Note that the optimal backwards twist is proportional to the ratio of a "backwards message" $p_{\boldsymbol{\theta}}(\mathbf{x}_t \mid \mathbf{y}_{t+1:T})$ and the latent marginal $p_{\boldsymbol{\theta}}(\mathbf{x}_t)$ (Equation 5). Thus, we can learn the backwards twist using density ratio estimation (DRE) [23, 24].

DRE via classification estimates the ratio of two densities $a(x)/b(x)$ by training a classifier to distinguish between samples from $a$ and $b$. If such a classifier is trained using the logit link function, then its raw output will approximate $\log a(x) - \log b(x)$ up to a constant [24]. Using this approach, we interpret $\log r_{\boldsymbol{\psi}}(\mathbf{y}_{t+1:T}, \mathbf{x}_t)$ as the logit of a Bernoulli classifier, which is trained to distinguish between samples from $p_{\boldsymbol{\theta}}(\mathbf{x}_t, \mathbf{y}_{t+1:T})$ and $p_{\boldsymbol{\theta}}(\mathbf{x}_t) p_{\boldsymbol{\theta}}(\mathbf{y}_{t+1:T})$, which are available from the model. When trained in this way, $\log r_{\boldsymbol{\psi}}(\mathbf{y}_{t+1:T}, \mathbf{x}_t)$ will approximate $\log p_{\boldsymbol{\theta}}(\mathbf{x}_t \mid \mathbf{y}_{t+1:T}) - \log p_{\boldsymbol{\theta}}(\mathbf{x}_t)$ up to a constant, which can be ignored. For details see Appendix C.2 and Sugiyama et al. [24].

We use the DRE-learned twisting functions in an alternating scheme that first holds $p_{\boldsymbol{\theta}}, q_{\boldsymbol{\theta}}$ fixed and updates $r_{\boldsymbol{\psi}}$ using density ratio estimation, and then holds $r_{\boldsymbol{\psi}}$ fixed and updates $p_{\boldsymbol{\theta}}$ and $q_{\boldsymbol{\theta}}$ by ascending a biased gradient estimator (no resampling terms) of $\mathcal{L}_{\text{SIXO}}(\boldsymbol{\theta}, \boldsymbol{\psi})$ in $\boldsymbol{\theta}$. We call the full alternating procedure for learning $\boldsymbol{\theta}$ and $\boldsymbol{\psi}$ SIXO-DRE, see Algorithm 1.

### 3.3 The SIXO Bound Can Become Tight

Maddison et al. [8] show that the FIVO bound can only become tight in models with uncommon dependency structures. We show that the SIXO bound can become tight for any model in the class defined in Section 2.

**Proposition 1.** Sharpness of the SIXO bound. *Let $p(\mathbf{x}_{1:T}, \mathbf{y}_{1:T})$ be a latent variable model with Markovian structure as defined in Section 2, let $\mathcal{Q}$ be the set of possible sequences of proposal distributions indexed by parameters $\boldsymbol{\theta} \in \Theta$, and let $\mathcal{R}$ be the set of possible sequences of positive, integrable twist functions indexed by parameters $\boldsymbol{\psi} \in \Psi$. Assume that $\{p(\mathbf{x}_t \mid \mathbf{x}_{t-1}, \mathbf{y}_{1:T})\}_{t=1}^{T} \in \mathcal{Q}$ and $\{p(\mathbf{y}_{t+1:T} \mid \mathbf{x}_t)\}_{t=1}^{T-1} \in \mathcal{R}$. Finally, assume $\mathcal{L}_{\text{SIXO}}(\boldsymbol{\theta}, \boldsymbol{\psi}, \mathbf{y}_{1:T})$ has the unique optimizer $\boldsymbol{\theta}^*, \boldsymbol{\psi}^* = \arg\max_{\boldsymbol{\theta} \in \Theta, \boldsymbol{\psi} \in \Psi} \mathcal{L}_{\text{SIXO}}(\boldsymbol{\theta}, \boldsymbol{\psi}, \mathbf{y}_{1:T})$.*

*Then the following holds:*

1. *$q_{\boldsymbol{\theta}^*}(\mathbf{x}_t \mid \mathbf{x}_{1:t-1}, \mathbf{y}_{1:T}) = p(\mathbf{x}_t \mid \mathbf{x}_{1:t-1}, \mathbf{y}_{1:T})$ for $t = 1, \ldots, T$,*

2. *$r_{\boldsymbol{\psi}^*}(\mathbf{y}_{t+1:T}, \mathbf{x}_t) \propto p(\mathbf{y}_{t+1:T} \mid \mathbf{x}_t)$ up to a constant independent of $\mathbf{x}_t$ for $t = 1, \ldots, T-1$,*

3. *$\mathcal{L}_{\text{SIXO}}^{K}(\boldsymbol{\theta}^*, \boldsymbol{\psi}^*, \mathbf{y}_{1:T}) = \log p(\mathbf{y}_{1:T})$ for any number of particles $K \geq 1$.*

*Proof.* See Appendix C.5. □

This is an important advantage of our work—the SIXO objective is the first to recover the true marginal likelihood with a finite number of particles while also being tailored to sequential tasks.

## 4 Related Work

Standard references for SMC include Doucet and Johansen [1], Naesseth et al. [2], and Del Moral [3] which provides a theoretical treatment of a generalization of SMC called Feynman-Kac formulae. Pitt and Shephard [17] introduced the auxiliary particle filter, an early smoothing SMC method which constructs an estimate of the one-step backwards message $p_{\boldsymbol{\theta}}(\mathbf{y}_{t+1} \mid \mathbf{x}_t)$ using simulations from the model. Smoothing SMC in general is discussed thoroughly in Briers et al. [15] and Del Moral et al. [25]. Later, Whiteley and Lee [13] introduced twisted particle filters, which perform SMC on a twisted model that approximates the smoothing distributions by multiplying the model's filtering distributions with "twisting" functions. Our work extends the theoretical framework in Whiteley and Lee [13] by proposing practical and effective methods for learning parametric twisting functions.

To make smoothing SMC computationally tractable, fixed-lag techniques use information from only a fixed window of future observations, as introduced in Clapp and Godsill [19] and surveyed in Lin et al. [20]. For example, Park and Ionides [26] use simulations from the model to estimate fixed-lag twisting functions and Doucet et al. [27] sample blocks of latents conditional on their observations via various Monte Carlo methods. These methods suffer from computational complexity that grows with the window size, and fail to take advantage of all future observations.

Other methods use twisting functions which depend on all observations. Most similar to our approach are Guarniero et al. [16], Heng et al. [28], and Zimmermann et al. [29] which learn parametric twists using a Bellman-like decomposition of the lookahead distributions $p(\mathbf{y}_{t+1:T} \mid \mathbf{x}_t)$ in terms of the same distributions one step into the future. Our method instead uses independent DRE objectives for the twist at each timestep. Del Moral and Murray [30] use Gaussian processes to approximate the twists, Lindsten et al. [18] use traditional graphical model techniques such as loopy belief propagation and expectation propagation, and Ruiz and Kappen [31] use optimal control techniques. None of these approaches consider model learning and their twist parameterizations and learning techniques are highly specialized to specific problem settings.

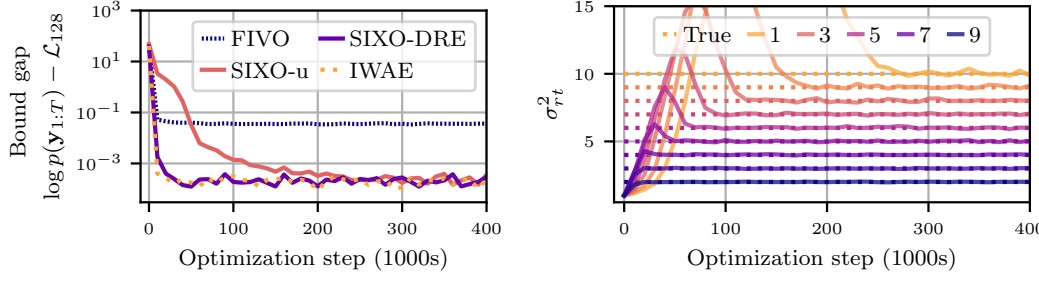

**(a)** Bound gap for different methods.

**(b)** Convergence of twist parameters with SIXO-u.

**Figure 2:** Results for the Gaussian drift diffusion experiment presented in Section 5.1. The median across ten random seeds are shown. Further figures and discussion are included in Appendix D.1.

Fitting model parameters via stochastic gradient ascent on an evidence lower bound (ELBO) was introduced in Ranganath et al. [32], Hoffman et al. [33], Kingma and Welling [34], and later generalized to the Monte Carlo objectives (MCO) framework by Mnih and Rezende [14]. Since then, works have considered optimizing lower bounds defined by the normalizing constant estimators from multiple importance sampling [35], nested importance sampling, [36, 29], rejection sampling and Hamiltonian Monte Carlo [12], filtering SMC [8–10], and smoothing SMC [11, 37, 38]. The prior work on smoothing SMC used an objective defined by forward filtering backwards smoothing [15] which suffers from the same particle degeneracy issues as filtering SMC and cannot become tight. Kim et al. [39] optimize the importance weighted autoencoder (IWAE) bound [35] using a *baseline* derived from future likelihood estimates, but do not use SMC or resampling in their bound.

## 5 Experiments

We experimentally explore our claims that:

1. The SIXO bound can become tight while FIVO cannot.
2. DRE-learned twists enable better posterior inference than filtering SMC.
3. Model learning with SIXO provides better parameter estimates than FIVO.

### 5.1 Gaussian Drift Diffusion

We first consider a one-dimensional Gaussian drift diffusion process with joint distribution

$$p_{\boldsymbol{\theta}}\left(\mathbf{x}_{1:T}, y_T\right) = \mathcal{N}\left(y_T \mid x_T + \alpha, \sigma_y^2\right) \mathcal{N}\left(x_1; \alpha, \sigma_x^2\right) \prod_{t=2}^{T} \mathcal{N}\left(x_t \mid x_{t-1} + \alpha, \sigma_x^2\right). \tag{6}$$

The single free model parameter is the drift $\alpha \in \mathbb{R}$, the state is $x_t \in \mathbb{R}$, and the observation is $y_T \in \mathbb{R}$. Figures 1a and 1b show that for $\alpha = 0$ the filtering and smoothing distributions in this model quickly diverge, which can lead to poor inference for filtering methods.

We compare joint model, proposal and twist learning using two variants of SIXO to variational inference with the IWAE bound [35] and FIVO with unbiased gradients [8–10]. All methods use an independent proposal at each time step parameterized as $q_t(x_t \mid x_{t-1}, y_T) = \mathcal{N}(x_t; f_t(x_{t-1}, y_T), \sigma_{qt}^2)$ where $f_t$ is an affine function, a family which contains the optimal proposal. SIXO-u uses twists parameterized as $r_t(y_T, x_t) = \mathcal{N}(y_T; g_t(x_t), \sigma_{rt}^2)$ where $g_t$ is an affine function, a family which contains the true lookahead distributions. The SIXO-DRE twist functions, $\log r_t(y_T, x_t)$, are parameterized as quadratic functions of $x_t$, where the parameters of the quadratic function are generated by a neural network with inputs $(y_T, t)$. The true log density ratio will be quadratic in $x_t$, so if the neural network is sufficiently flexible, the true log density ratio function can be obtained.

Figure 2a shows the convergence of the variational bound for each method. As expected IWAE recovers a tight variational bound, whereas FIVO does not. While SIXO-u does recover a tight variational bound, the high variance of the unbiased gradient estimator makes it slower to converge,

**Table 1:** Performance of FIVO and SIXO on the SVM.

| Method | Train $\mathcal{L}^4_{\text{Method}}$ (as in [9]) | Train $\mathcal{L}^{2048}_{\text{BPF}}$ | Test $\mathcal{L}^{2048}_{\text{BPF}}$ |
|---|---|---|---|
| IWAE | $\mathbf{6940.18 \pm 1.17}$ | $\mathbf{7019.38 \pm 2.99}$ | $3351.21 \pm 2.35$ |
| FIVO | $6921.29 \pm 1.33$ | $\mathbf{7020.14 \pm 2.86}$ | $3352.51 \pm 1.30$ |
| SIXO-q | $6928.90 \pm 1.24$ | $\mathbf{7019.65 \pm 2.97}$ | $3353.10 \pm 1.58$ |
| SIXO-DRE | $6931.51 \pm 2.08$ | $\mathbf{7019.42 \pm 3.01}$ | $\mathbf{3354.08 \pm 1.60}$ |

and impractical for non-toy problems. Conversely, SIXO-DRE achieves a tight bound while using the lower variance biased gradients. This motivates its use in more complex, non-linear settings where the unbiased FIVO gradients are not practical. Figure 2b shows that SIXO-u recovers the correct twist parameters. More figures illustrating the convergence of $\boldsymbol{\theta}$ and $\psi$ are included in Appendix D.1.

In Figures 1c and 1d we compare particle trajectories under FIVO and SIXO-u. We see that FIVO consistently proposes particles with high likelihood under the posterior distributions (identical to the smoothing distributions in this case) which are discarded by the resampling steps in filtering SMC. In contrast, SIXO both proposes particles with high posterior likelihood and retains them through the resampling steps by properly scoring particles under the twisted target distributions. These results empirically verify the theoretical claims made in Section 3.3.

## 5.2 Stochastic Volatility Model

We now apply SIXO to a stochastic volatility model (SVM) of monthly foreign exchange rates for $N = 22$ currencies, over the period 9/2007 to 8/2017 [40]. The SVM generative model is defined as

$$\mathbf{x}_1 \sim \mathcal{N}(\mathbf{0}, \mathbf{Q}), \quad \mathbf{x}_t = \boldsymbol{\mu} + \boldsymbol{\phi} \odot (\mathbf{x}_{t-1} - \boldsymbol{\mu}) + \boldsymbol{\nu}_t, \quad \mathbf{y}_t = \boldsymbol{\beta} \odot \exp\left(\frac{\mathbf{x}_t}{2}\right) \odot \boldsymbol{\varepsilon}_t, \quad (7)$$

with transition noise $\boldsymbol{\nu}_t \sim \mathcal{N}(\mathbf{0}, \mathbf{Q})$, observation noise $\boldsymbol{\varepsilon}_t \sim \mathcal{N}(\mathbf{0}, I_{N \times N})$, states $\mathbf{x}_{1:T} \in \mathbb{R}^{T \times N}$, and observations $\mathbf{y}_{1:T} \in \mathbb{R}^{T \times N}$. All multiplications are performed element-wise, denoted by $\odot$. The proposal, $q_{\boldsymbol{\theta}}$, is structured as $q_{\boldsymbol{\theta}}(\mathbf{x}_{1:T}) \propto \prod_{t=1}^{T} \mathcal{N}(\mathbf{x}_t; \boldsymbol{\mu}_t, \boldsymbol{\Sigma}_t) p_{\boldsymbol{\theta}}(\mathbf{x}_t \mid \mathbf{x}_{t-1})$ with means $\boldsymbol{\mu}_t \in \mathbb{R}^N$ and diagonal covariance matrices $\boldsymbol{\Sigma}_t \in \text{diag}(\mathbb{R}^N_+)$. We compare four approaches: IWAE, FIVO, SIXO with quadrature twist (SIXO-q), and SIXO with density ratio twist (SIXO-DRE). Note that the observations for this model are dense in time, so we would expect filtering-based approaches to perform well. See Appendix D.2 for more details.

**Train Performance** We first compare our methods in terms of their 4-particle log marginal likelihood lower bounds as in Naesseth et al. [9], e.g. for IWAE we report the IWAE bound and for FIVO we report the FIVO bound. Even though SIXO-q only scores a single future observation, it still obtains a 7-nat improvement over FIVO. SIXO-DRE, meanwhile, conditions on all future observations and obtains a 10-nat improvement over FIVO. All SMC-based methods, however, significantly underperform IWAE which seems to indicate that IWAE's unbiased gradients and use of smoothing information allow it to learn a proposal which is efficient at low numbers of particles. This performance does not carry over to model learning or other high-dimensional problems, however, and is at odds with previous work on this model and dataset [8–10].

We also estimate the learned model's training set marginal likelihood by computing a bootstrap particle filter's (BPF) log marginal lower bound with 2,048 particles, denoted $\mathcal{L}^{2048}_{\text{BPF}}$ [41]. Interestingly, a one-way ANOVA [42] does not reject the null hypothesis that the training set $\mathcal{L}^{2048}_{\text{BPF}}$ means are all equal ($p = 0.26$), suggesting that the log marginal likelihoods on training data are indistinguishable and training performance has saturated. It is clear, however, that among the SMC methods SIXO performs the best inference and makes the most efficient use of particles as the $\mathcal{L}^4_{\text{SIXO}}$ bounds are significantly higher than $\mathcal{L}^4_{\text{FIVO}}$ for similar models.

**Test Performance** We also compare methods on a held-out test set to evaluate model learning. We construct this test set using the same data source as the training set, but use the period of time since Naesseth et al. [9] was published, an extra 55 months. Again, we report BPF log marginal lower bounds with 2,048 particles and find that SIXO-DRE outperforms IWAE, SIXO-q and FIVO. All differences are statistically significant.

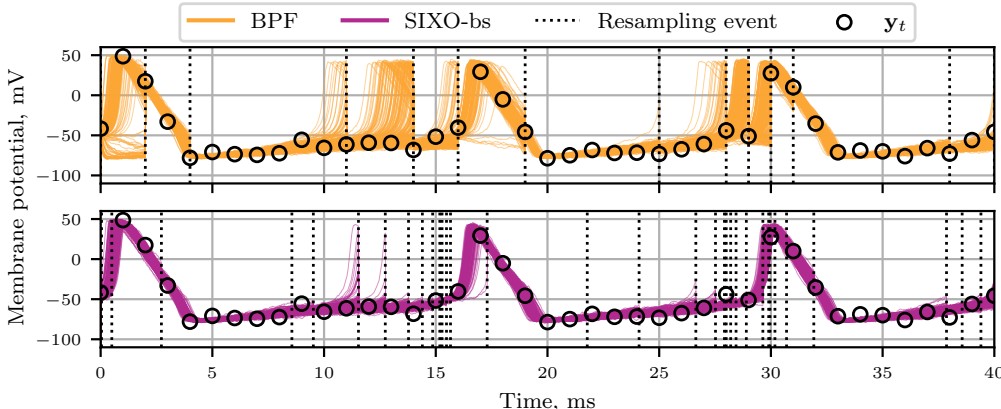

**Figure 3:** Comparison of trajectories generated by a BPF (top) and a SIXO-bs sweep (bottom) on synthetic data from the Hodgkin-Huxley model. Both sweeps use the true model parameters and a bootstrap proposal while SIXO-bs uses a learned twist. SIXO-bs resamples more frequently because the twist changes the SMC weights at each model step, while the BPF only changes weights at each observation. This allows SIXO-bs to resample away erroneous spikes right as they begin.

**Table 2:** Hodgkin-Huxley inference performance for different observation intervals.

| Observation Interval | $\mathcal{L}^{32}_{\mathrm{Method}}$ / number of observations | | | |
|:---:|:---:|:---:|:---:|:---:|
| | SIXO-sm | SIXO-bs | FIVO-fi | FIVO-bs |
| 2ms | $-1.66 \pm 0.054$ | $-1.75 \pm 0.100$ | $-22.80 \pm 0.457$ | $-2.57 \pm 0.297$ |
| 1ms | $-1.18 \pm 0.004$ | $-1.36 \pm 0.206$ | $-11.86 \pm 0.304$ | $-2.36 \pm 0.356$ |
| 0.5ms | $-1.06 \pm 0.013$ | $-1.17 \pm 0.254$ | $-6.09 \pm 0.302$ | $-2.17 \pm 0.305$ |
| 0.2ms | $-0.92 \pm 0.003$ | $-1.00 \pm 0.209$ | $-2.49 \pm 0.106$ | $-1.93 \pm 0.243$ |

## 5.3 Hodgkin-Huxley Model

We conclude by comparing FIVO and SIXO on the Hodgkin-Huxley (HH) model of neural action potentials [43, 44]. A single neuron is represented with a four-dimensional state-space: the instantaneous membrane potential and the relative conductivity of three ion gates. A noise-corrupted and subsampled membrane potential can be obtained using electrodes [45] or voltage imaging [46]. The state of the gates, however, is not observable, and must be inferred from the noisy potential recordings. The physiological parameters governing the time-evolution of the system are also of interest, such as the base conductance of each of the ion channels.

We implement the HH model as a four-dimensional nonlinear state space model with Gaussian transition noise [47]. The observation is a single Gaussian-distributed value with mean equal to the instantaneous potential. Unless otherwise stated, we subsample observations by a factor of 50 to simulate an acquisition frequency of 1kHz (interval of 1ms). For more details, see Appendix D.3.

In this model action potentials, or spikes, are rare events that happen quickly and invoke a rapid change in the state. Therefore, filtering-based inference is particularly disadvantageous as noisy observations may trigger erroneous spikes or "miss" true spikes.

We compare four methods: SIXO with a DRE twist and a learned smoothing proposal (SIXO-sm), SIXO with a bootstrap proposal (SIXO-bs), FIVO with a learned filtering proposal (SIXO-fi), and FIVO with a bootstrap proposal (FIVO-bs).

**Inference** In Figure 3 we compare HH trajectories generated by a BPF and SIXO-bs. The BPF yields spurious spikes and misses the initiation of other spikes, illustrating the issue with filtering SMC for HH inference. SIXO-bs allows fewer spurious spikes to develop and fewer particles miss the onset of spikes. SIXO-bs also achieves a log marginal lower bound of $-47.81$ nats, higher than the $-49.24$ nats achieved by the BPF, showing that it performs more effective inference. In Table 2 we

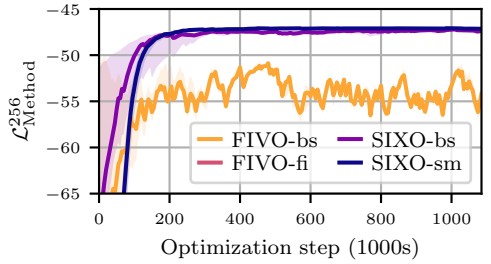
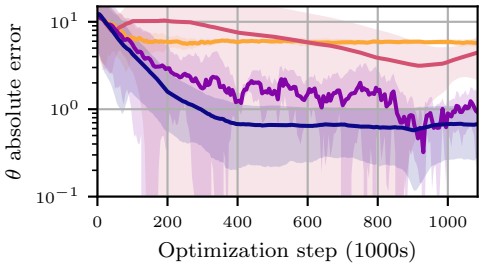

**(a)** Validation set $\mathcal{L}^{256}_{\mathrm{Method}}$ over training.      **(b)** Relative parameter error over training.

**Figure 4:** Hodgkin-Huxley model learning performance over training. Both SIXO-bs and SIXO-sm obtain a better parameter estimate and more stable bound than FIVO. FIVO-fi is not stable and achieves a low bound, not visible on Figure 4a.

**Table 3:** Hodgkin-Huxley model learning performance.

| Bound | Proposal | Test $\mathcal{L}^{256}_{\mathrm{Method}}$ | Test $\mathcal{L}^{256}_{\mathrm{BPF}}$ | $\theta$ Relative Error |
|---|---|---|---|---|
| (True model) | (Bootstrap) | (N/A) | $(-49.12)$ | $(0.0)$ |
| FIVO | Bootstrap | $-51.32 \pm 0.73$ | $-51.32 \pm 0.73$ | $0.45 \pm 0.03$ |
| FIVO | Filtering | $-660.62 \pm 283.50$ | $-115.74 \pm 80.12$ | $0.78 \pm 0.20$ |
| SIXO | Bootstrap | $-48.98 \pm 0.20$ | $-49.40 \pm 0.79$ | $0.10 \pm 0.06$ |
| **SIXO** | **Smoothing** | $\mathbf{-48.73 \pm 0.14}$ | $\mathbf{-49.30 \pm 0.71}$ | $\mathbf{0.05 \pm 0.02}$ |

study this effect more extensively, comparing performance across different observation frequencies. We see, as predicted, that the performance of SIXO is more consistent across observation frequencies than FIVO, supporting the claim that twists assist in inference when observations are sparse. For more results, see Table 4 in Appendix D.3.

**Model Learning**    We conclude by comparing FIVO and SIXO for parameter recovery in Figure 4 and Table 3. We see that FIVO-bs converges to a poor parameter estimate, and recovers a poor variational bound. FIVO-fi does not converge (details and full figure in Appendix D.3). The SIXO methods recover much better parameter estimates, and achieve the highest bound values, with SIXO-sm outperforming all other methods in terms of final performance and training stability.

## 6   Conclusions, Limitations, and Future Work

In this work we proposed a method of learning twisting functions for smoothing SMC via density ratio estimation. Our approach ascends a lower bound on the log marginal likelihood that can theoretically become tight, a first for SMC objectives. We verified our theoretical claims by experimentally demonstrating improvements over existing techniques in inference and model learning.

There are, however, important limitations to our approach. Training and evaluating the twist requires additional computational effort compared to FIVO, and requires further hyperparameter tuning. Although we consider SIXO to be mainly used for offline settings, extending SIXO to online settings could yield practical benefit. Finally, there is a known pathology in DRE methods where the ratio may be poorly estimated if the difference between the densities in the ratio is very large [48]. Thus, new methods for learning the twist are important topics for future work.

**Acknowledgements and Disclosure of Funding**    We thank Matt MacKay for helpful discussions and edits, and the anonymous reviewers for their feedback during the review process. This work was supported by grants from the Simons Collaboration on the Global Brain (SCGB 697092), the NIH BRAIN Initiative (U19NS113201 and R01NS113119). Some of the computation for this work was made possible by Microsoft Education Azure cloud credits. Dieterich Lawson was supported in part by a Stanford Data Science Fellowship.

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
