# Appendices for SIXO: Smoothing Inference with Twisted Objectives

## A    Table of Notation

| Name | Symbol | Notes |
| --- | --- | --- |
| Sequence length | $T$ | $T \in \mathbb{N}$ |
| Timestep | $t$ | $t \in \{1, \dots, T\}$ |
| Latent state | $\mathbf{x}_t$ | $\mathbf{x}_t \in \mathcal{X}$ |
| Observation | $\mathbf{y}_t$ | $\mathbf{y}_t \in \mathcal{Y}$ |
| Observation sequence | $\mathbf{y}_{1:T}$ | $\mathbf{y}_{1:T} \in \mathcal{Y}^T$ |
| Number of SMC particles | $K$ | $K \in \mathbb{N}$ |
| $k^{\text{th}}$ particle latent trajectory | $\mathbf{x}_{1:t}^k$ | $\mathbf{x}_{1:t}^k \in \mathcal{X}^t$ |
| Model and proposal parameters | $\boldsymbol{\theta}$ | $\boldsymbol{\theta} \in \boldsymbol{\Theta}$ |
| Twist parameters | $\boldsymbol{\psi}$ | $\boldsymbol{\psi} \in \boldsymbol{\Psi}$ |
| Joint distribution | $p_{\boldsymbol{\theta}}(\mathbf{x}_{1:T}, \mathbf{y}_{1:T})$ | Distribution on $\mathcal{X}^T \times \mathcal{Y}^T$ |
| $t^{\text{th}}$ transition distribution | $p_{\boldsymbol{\theta}}(\mathbf{x}_t \mid \mathbf{x}_{t-1})$ | Conditional distribution on $\mathcal{X}$ |
| $t^{\text{th}}$ observation distribution | $p_{\boldsymbol{\theta}}(\mathbf{y}_t \mid \mathbf{x}_t)$ | Conditional distribution on $\mathcal{Y}$ |
| Proposal distributions | $\{q_{\boldsymbol{\theta}}(\mathbf{x}_t \mid \mathbf{x}_{t-1}, \mathbf{y}_{1:T})\}_{t=1}^T$ | $T$ conditional distributions on $\mathcal{X}$ |
| Twist functions | $\{r_{\boldsymbol{\psi}}(\mathbf{y}_{t+1:T}, \mathbf{x}_t)\}_{t=1}^{T-1}$ | $T-1$ positive integrable functions in $\mathcal{X} \times \mathcal{Y}^{T-t-1} \times \boldsymbol{\Psi} \to \mathbb{R}_{\geq 0}$ |
| Filtering distributions | $\{p_{\boldsymbol{\theta}}(\mathbf{x}_{1:t} \mid \mathbf{y}_{1:t})\}_{t=1}^T$ | $T$ conditional distributions on $\mathcal{X}^t$ |
| Smoothing distribution | $\{p_{\boldsymbol{\theta}}(\mathbf{x}_{1:t} \mid \mathbf{y}_{1:T})\}_{t=1}^T$ | $T$ conditional distributions on $\mathcal{X}^t$ |
| Unnormalized target distributions | $\{\gamma_t(\mathbf{x}_{1:t})\}_{t=1}^T$ | $T$ positive integrable functions in $\mathcal{X}^t \to \mathbb{R}_{\geq 0}$ |
| Normalized target distributions | $\{\pi_t(\mathbf{x}_{1:t})\}_{t=1}^T$ | $T$ distributions on $\mathcal{X}^t$ |
| $t^{\text{th}}$ normalizing constant | $Z_t$ | Positive real, $\pi_t(\mathbf{x}_{1:t}) = \gamma_t(\mathbf{x}_{1:t})/Z_t$ |
| Bootstrap particle filter | BPF | SMC using $p(\mathbf{x}_t \mid \mathbf{x}_{t-1})$ as the proposal and no twist. |
| Filtering sequential Monte Carlo | FSMC | SMC with filtering distributions as targets. |
| SIXO-unified | SIXO-u | SIXO objective optimized w.r.t. $\boldsymbol{\theta}$ and $\boldsymbol{\psi}$ through direct and unbiased gradient ascent of the SIXO bound. |
| SIXO-quadrature | SIXO-q | SIXO objective with $r_t$ parameterized using the quadrature twist, and optimized using ascent of the unified SIXO bound. |
| SIXO-density ratio estimation | SIXO-DRE | SIXO with $r_t$ parameterized as the density ratio estimate twist, and learned using the alternating method. |
| SIXO with DRE and bootstrap proposal | SIXO-bs | Used in HH experiments. |
| SIXO with DRE and learned smoothing proposal | SIXO-sm | Used in HH experiments. |
| FIVO with bootstrap proposal | FIVO-bs | Used in HH experiments. Same forward sweep as a BPF. |
| FIVO with learned filtering proposal | FIVO-fi | Used in HH experiments. |
| BPF bound | $\mathcal{L}_{\text{BPF}}^K$ | Log marginal likelihood bound from a $K$-particle BPF. |
| FIVO bound | $\mathcal{L}_{\text{FIVO}}^K$ | FIVO bound (2) with $K$ particles. |
| SIXO bound | $\mathcal{L}_{\text{SIXO}}^K$ | SIXO bound (3) with $K$ particles. |
| DRE loss | $\mathcal{L}_{\text{DRE}}$ | Loss used to learn twist with DRE. |

# B Background

## B.1 Sequential Monte Carlo

Sequential Monte Carlo (SMC) is a popular method for sampling from posterior distributions with sequential structure. For a thorough introduction we refer the reader to Doucet and Johansen [1] and Naesseth et al. [2]. We reproduce the general SMC algorithm in Algorithm 2.

---

**Algorithm 2** Sequential Monte Carlo

---

1: **procedure** SMC($\{\gamma_t(\mathbf{x}_{1:t})\}_{t=1}^T, \{q_t(\mathbf{x}_t \mid \mathbf{x}_{1:t-1})\}_{t=1}^T, K$)
2: $\quad w_0^{1:K} = 1, \quad \widehat{Z}_0 = 1$
3: $\quad$ **for** $t = 1, \ldots, T$ **do**
4: $\quad\quad$ **for** $k = 1, \ldots, K$ **do**
5: $\quad\quad\quad \mathbf{x}_t^k \sim q_t(\mathbf{x}_t | \mathbf{x}_{1:t-1}^k)$
6: $\quad\quad\quad \mathbf{x}_{1:t}^k = (\mathbf{x}_{1:t-1}^k, \mathbf{x}_t^k)$
7: $\quad\quad\quad \alpha_t^k = \dfrac{\gamma_t(\mathbf{x}_{1:t}^k)}{\gamma_t(\mathbf{x}_{1:t-1}^k)q_t(\mathbf{x}_t^k \mid \mathbf{x}_{1:t-1}^k)}$
8: $\quad\quad\quad w_t^k = w_{t-1}^k \alpha_t^k$
9: $\quad\quad \widehat{Z_t/Z_{t-1}} = \dfrac{\sum_{k=1}^K w_t^k}{\sum_{k=1}^K w_{t-1}^k}$
10: $\quad\quad \widehat{Z}_t = \widehat{Z}_{t-1}(\widehat{Z_t/Z_{t-1}})$
11: $\quad\quad$ **if** should resample **then**
12: $\quad\quad\quad$ **for** $k = 1, \ldots, K$ **do**
13: $\quad\quad\quad\quad a_t^k \sim \text{Categorical}(w_t^{1:K})$
14: $\quad\quad\quad\quad \tilde{\mathbf{x}}_{1:t}^k = \mathbf{x}_{1:t}^{a_t^k}$
15: $\quad\quad\quad \mathbf{x}_{1:t}^{1:K} = \tilde{\mathbf{x}}_{1:t}^{1:K}$
16: $\quad\quad\quad w_t^{1:K} = 1$
17: $\quad$ **return** $\widehat{Z}_T, \mathbf{x}_{1:T}^{1:K}$

---

## B.2 Factoring the Smoothing Distributions

Here we show that the smoothing distributions can be factored into the filtering distributions and the lookahead distributions.

Let $p_{\boldsymbol{\theta}}(\mathbf{x}_{1:T}, \mathbf{y}_{1:T})$ be a model as defined in (1). Then

$$p_{\boldsymbol{\theta}}(\mathbf{x}_{1:t}, \mathbf{y}_{1:T}) = p_{\boldsymbol{\theta}}(\mathbf{x}_{1:t}, \mathbf{y}_{1:t})p_{\boldsymbol{\theta}}(\mathbf{y}_{t+1:T} \mid \mathbf{y}_{1:t}, \mathbf{x}_{1:t}) \tag{8}$$

$$= p_{\boldsymbol{\theta}}(\mathbf{x}_{1:t}, \mathbf{y}_{1:t})p_{\boldsymbol{\theta}}(\mathbf{y}_{t+1:T} \mid \mathbf{x}_t), \tag{9}$$

where $\mathbf{y}_{t+1:T}$ is conditionally independent of $(\mathbf{y}_{1:t}, \mathbf{x}_{1:t-1})$ given $\mathbf{x}_t$ because of the Markov structure of $p_{\boldsymbol{\theta}}$.

# C Methods

## C.1 The Gradients of SMC

The original FIVO papers [8–10] used biased gradients to optimize $\mathcal{L}_{\text{FSMC}}$ which ignore score-function gradient terms arising from the discrete resampling operations. We use the same biased gradient estimator in SIXO-DRE when optimizing $\mathcal{L}_{\text{SIXO}}(\boldsymbol{\theta}, \boldsymbol{\psi}, \mathbf{y}_{1:T})$ in terms of $\boldsymbol{\theta}$. We also examine SIXO-u, where we use the full unbiased gradient estimator (see (17)). We derive both estimators here.

Assume resampling occurs at each timestep, let $\mathbf{A}_t = a_t^{1:K}$ and $\mathbf{X}_t = \mathbf{x}_t^{1:K}$ be the ancestor indices and latents for all particles at time $t$, and let $\mathbf{A} = (\mathbf{A}_1, \ldots, \mathbf{A}_{T-1})$ and $\mathbf{X} = (\mathbf{X}_1, \ldots, \mathbf{X}_T)$ be the

full sequence of ancestor indices and latents. We can then write the probability distribution over $\mathbf{X}, \mathbf{A}$ that defines SMC as

$$p_{\mathrm{SMC}}(\mathbf{X}, \mathbf{A}) = \left(\prod_{k=1}^{K} q(\mathbf{x}_1^k)\right) \prod_{t=2}^{T} \prod_{k=1}^{K} q(\mathbf{x}_t^k | \mathbf{x}_{1:t-1}^{a_{t-1}^k}) \overline{\alpha}_{t-1}^{a_{t-1}^k} \tag{10}$$

where $\overline{\alpha}_t^i = \alpha_t^i / (\sum_{k=1}^{K} \alpha_t^k)$ is the normalized incremental weight. When the proposals and targets of SMC are parametric functions of $\boldsymbol{\theta}$, both $q$ and $\overline{\alpha}$ will depend on $\boldsymbol{\theta}$.

To emphasize its dependence on $\boldsymbol{\theta}$ when run with parametric twists and proposals we will write $p_{\mathrm{SMC}}(\mathbf{X}, \mathbf{A})$ as $p(\mathbf{X}, \mathbf{A}; \boldsymbol{\theta})$. Then the gradient of $\mathcal{L}_{\mathrm{SIXO}}$ is defined as

$$\nabla_{\boldsymbol{\theta}}(\mathcal{L}_{\mathrm{SIXO}}(\boldsymbol{\theta})) = \nabla_{\boldsymbol{\theta}} \mathbb{E}_{\mathbf{X}, \mathbf{A} \sim p(\mathbf{X}, \mathbf{A}; \boldsymbol{\theta})} [\log \widehat{Z}(\mathbf{X}, \mathbf{A}, \boldsymbol{\theta})] \tag{11}$$

where we have rewritten $\widehat{Z}_{\mathrm{SIXO}}(\boldsymbol{\theta}, \boldsymbol{\psi}, \mathbf{y}_{1:T})$ as $\widehat{Z}(\mathbf{X}, \mathbf{A}, \boldsymbol{\theta})$ to emphasize its dependence on the random variables $\mathbf{X}$ and $\mathbf{A}$ and suppress its dependence on $\boldsymbol{\psi}$ and $\mathbf{y}_{1:T}$.

The first step is to reparameterize the expectation in terms of continuous noise instead of $\mathbf{X}$ [49, 50, 34, 51, 52]. Assume $\mathbf{X}$ is from a reparameterizable distribution and let $\phi_{\mathbf{X}}(\boldsymbol{\theta}, \epsilon)$ be a function that deterministically combines continuous noise $\epsilon$ and the parameters $\boldsymbol{\theta}$ to produce a sample $\mathbf{X}$. Then we have

$$\nabla_{\boldsymbol{\theta}} \mathbb{E}_{\mathbf{X}, \mathbf{A}}[\log \widehat{Z}(\mathbf{X}, \mathbf{A}, \boldsymbol{\theta})] = \nabla_{\boldsymbol{\theta}} \mathbb{E}_{\epsilon, \mathbf{A}}[\log \widehat{Z}(\phi_{\mathbf{X}}(\boldsymbol{\theta}, \epsilon), \mathbf{A}, \boldsymbol{\theta})]. \tag{12}$$

We will further abuse notation by writing $\widehat{Z}(\phi_{\mathbf{X}}(\boldsymbol{\theta}, \epsilon), \mathbf{A}, \boldsymbol{\theta})$ as $\widehat{Z}(\epsilon, \mathbf{A}, \boldsymbol{\theta})$. Rewriting the expectation (12) as an integral gives

$$\nabla_{\boldsymbol{\theta}} \mathbb{E}_{\epsilon, \mathbf{A}}[\log \hat{Z}(\epsilon, \mathbf{A}, \boldsymbol{\theta})] = \nabla_{\boldsymbol{\theta}} \int \log \hat{Z}(\epsilon, \mathbf{A}, \boldsymbol{\theta}) p(\epsilon, \mathbf{A}; \boldsymbol{\theta}) d\epsilon d\mathbf{A}. \tag{13}$$

Assuming that we can differentiate under the integral allows us to break the integrand apart using the product rule as

$$\int \nabla_{\boldsymbol{\theta}}(\log \widehat{Z}(\epsilon, \mathbf{A}, \boldsymbol{\theta})) p(\epsilon, \mathbf{A}; \boldsymbol{\theta}) + \log \widehat{Z}(\epsilon, \mathbf{A}, \boldsymbol{\theta}) \nabla_{\boldsymbol{\theta}}(p(\epsilon, \mathbf{A}; \boldsymbol{\theta})) d\epsilon d\mathbf{A}. \tag{14}$$

The left hand term in the integrand of (14) equals $\mathbb{E}_{\epsilon, \mathbf{A}}[\nabla_{\boldsymbol{\theta}} \log \widehat{Z}(\epsilon, \mathbf{A}, \boldsymbol{\theta})]$, the expectation of a gradient that can be estimated using simple Monte Carlo. The right hand term is a "score-function gradient" [53] which can be rewritten using the fact that $\nabla_{\boldsymbol{\theta}}(\log f(\boldsymbol{\theta})) = \nabla_{\boldsymbol{\theta}}(f(\boldsymbol{\theta}))/f(\boldsymbol{\theta})$ as

$$\int \log \widehat{Z}(\epsilon, \mathbf{A}, \boldsymbol{\theta}) \nabla_{\boldsymbol{\theta}}(\log p(\epsilon, \mathbf{A}; \boldsymbol{\theta})) p(\epsilon, \mathbf{A}; \boldsymbol{\theta}) d\epsilon d\mathbf{A} \tag{15}$$

which in turn equals the expectation

$$\mathbb{E}_{\epsilon, \mathbf{A}} \left[ \log \widehat{Z}(\epsilon, \mathbf{A}, \boldsymbol{\theta}) \nabla_{\boldsymbol{\theta}} \log p(\epsilon, \mathbf{A}; \boldsymbol{\theta}) \right]. \tag{16}$$

Writing both terms together gives the full unbiased gradient that is amenable to estimation with simple Monte Carlo,

$$\mathbb{E}_{\epsilon, \mathbf{A}} \left[ \nabla_{\boldsymbol{\theta}} \log \widehat{Z}(\epsilon, \mathbf{A}, \boldsymbol{\theta}) + \log \widehat{Z}(\epsilon, \mathbf{A}, \boldsymbol{\theta}) \nabla_{\boldsymbol{\theta}} \log p(\epsilon, \mathbf{A}; \boldsymbol{\theta}) \right]. \tag{17}$$

Similar to prior work [8–11] we find that the term on the right hand side of (17) has prohibitively high variance which inhibits learning. Dropping it gives the biased SMC gradient estimator used in SIXO-DRE,

$$\mathbb{E}_{\epsilon, \mathbf{A}}[\nabla_{\boldsymbol{\theta}} \log \widehat{Z}(\epsilon, \mathbf{A}, \boldsymbol{\theta})], \tag{18}$$

which can be estimated using open-source autodiff software [54].

The derivation above is adaptable for any resampling schedule that does not depend on the parameters (and by extension, the weights), but many common resampling schemes such as effective sample size resampling do not meet this requirement. If the resampling scheme depends on the parameters of the model, it introduces additional gradient terms which are not described here. Thus for all methods in experiments where compare to SIXO-u we use a fixed resampling schedule.

## C.2 Density Ratio Estimation

Density ratio estimation (DRE) considers estimating ratios of densities, e.g. $a(x)/b(x)$ with $a(x)$ and $b(x)$ defined on the same probability space and $b(x) > 0$ for all $x$. Instead of estimating $a(x)$ and $b(x)$ individually and then forming the ratio, an alternative approach is to directly estimate the *odds* that a given sample of $x$ was drawn from $a$.

Let $p(x, z) = p(z)p(x \mid z)$ be an expanded generative model for $x$ defined as

$$z \sim \text{Bernoulli}(\alpha), \tag{19}$$
$$x \sim a(x) \quad \text{if} \quad z = 1, \tag{20}$$
$$x \sim b(x) \quad \text{if} \quad z = 0 \tag{21}$$

with $\alpha \in (0, 1)$. We can now write the density ratio in terms of conditionals in this generative model,

$$a(x)/b(x) = p(x \mid z = 1) \,/\, p(x \mid z = 0) \tag{22}$$
$$= \left( \frac{p(x)p(z = 1 \mid x)}{p(z = 1)} \right) \Big/ \left( \frac{p(x)p(z = 0 \mid x)}{p(z = 0)} \right), \tag{23}$$
$$= \left( \frac{p(z = 0)}{p(z = 1)} \right) \Big/ \left( \frac{p(z = 0 \mid x)}{p(z = 1 \mid x)} \right), \tag{24}$$
$$= \left( \frac{1 - \alpha}{\alpha} \right) \left( \frac{p(z = 1 \mid x)}{p(z = 0 \mid x)} \right). \tag{25}$$

Thus, the density ratio can be rewritten as proportional to the odds that $x$ was drawn from $a(x)$ instead of $b(x)$.

**Density ratio estimation via classification** suggests training a binary classifier with supervised learning to predict $z$ given $x$ [24, 23]. Let $\sigma(x) = 1/(1 + e^{-x})$ be the sigmoid function and let $g_\psi(x)$ be a classifier trained with Bernoulli loss to maximize the log probability of a dataset $z_{1:N}, x_{1:N}$ sampled IID from $p(x, z)$. Specifically, $\psi$ is fit by minimizing $\mathcal{L}_{\text{DRE}}(\psi)$, defined as

$$\mathcal{L}_{\text{DRE}}(\psi) \triangleq \mathbb{E}_{z_{1:N}, x_{1:N} \sim p(x,z)} \left[ \log \prod_{i=1}^{N} \text{Bernoulli}(z; \sigma(g_\psi(x))) \right], \tag{26}$$

$$= \mathbb{E}_{z_{1:N}, x_{1:N} \sim p(x,z)} \left[ \sum_{i=1}^{N} z_i \log(\sigma(g_\psi(x_i))) + (1 - z_i) \log(\sigma(g_\psi(x_i))) \right]. \tag{27}$$

If trained in this way, the raw output of $g_\psi(x)$ will approximate the log-odds that $x$ came from $a(x)$ instead of $b(x)$, i.e.

$$g_\psi(x) \approx \log \left( \frac{p(z = 1 \mid x)}{1 - p(z = 1 \mid x)} \right) = \log \left( \frac{p(z = 1 \mid x)}{p(z = 0 \mid x)} \right). \tag{28}$$

The log of the density ratio can then be expressed as

$$\log a(x)/b(x) = \log(1 - \alpha) - \log(\alpha) + \log \left( \frac{p(z = 1 \mid x)}{p(z = 0 \mid x)} \right), \tag{29}$$
$$\approx \log(1 - \alpha) - \log(\alpha) + g_\psi(x). \tag{30}$$

Assuming a fixed $\alpha$ parameter, the log-ratio of densities is then proportional to the logit produced by $g_\psi$ up to an additive constant. Thus as long as we can sample training pairs $(x, z)$ from the expanded generative model above, we can estimate ratios of densities by training a binary classifier.

## C.3 Backwards Density Ratio Twist Architecture

We now provide further implementation details for the the backwards DRE twist we propose. The twist is defined through two neural networks, as illustrated in Figure 5. First, a recurrent neural network (RNN) is passed *backwards* over the data to create a sequence of encodings, denoted $\mathbf{e}_{2:T}$. These backward encodings are computed upfront, before the SMC sweep, with a cost complexity that

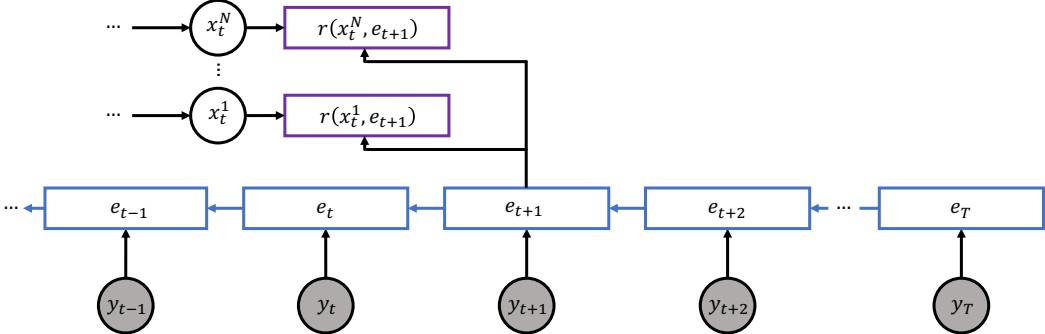

**Figure 5:** The backwards DRE twist architecture. Shown in blue is the recurrent backwards encoding, denoted here as $\mathbf{e}_t$, computed before the sweep. This computation will be shared across all particles. At time $t$, the twist value for each particle is computed by taking the backwards encoding at the next timestep, $\mathbf{e}_{t+1}$, concatenating it with the particle, and passing it through an MLP, shown in purple. As such, the twist complexity is linear in both time and the number of particles.

is linear in the length of the sequence. To evaluate the twist value of the $n^{\text{th}}$ particle at time $t$, denoted $r_t^n$, a second "head" MLP is used. The head MLP accepts as inputs the backwards encoding at the *next* timestep, $\mathbf{e}_{t+1}$, and the particle value, $\mathbf{x}_t^n$, and produces a single real-valued scalar, representing the logarithm of the twist value, $\log r_t^n$. The total cost of application of the head MLP is linear in time and the number of particles. As the head MLP reuses the backwards encoding, the cost of applying the backwards RNN is amortized across the number of particles used in the forward sweep. Note that no twist is computed at $t = T$, i.e. $r_T^k \triangleq 1$.

### C.4 Alternating Density Ratio Twist Training

To train the density ratio twist functions we define a supervised maximum likelihood update that is applied offline from any update to the model and proposal parameters. This update is shown in Algorithm 1, labeled as DRE.

To define this update, we use the approach in Section C.2 and set $a(x) = p_{\boldsymbol{\theta}}(\mathbf{x}_t \mid \mathbf{y}_{t+1:T})$ and $b(x) = p_{\boldsymbol{\theta}}(\mathbf{x}_t)$. To generate positive and negative examples for DRE, we first sample a set of $M$ latent state and observation trajectories from the generative model, $\mathbf{x}_{1:T}^{1:M}, \mathbf{y}_{1:T}^{1:M} \sim p_{\boldsymbol{\theta}}(\mathbf{x}_{1:T}, \mathbf{y}_{1:T})$. Because these are samples from joint distribution they are also samples from the conditionals $\{p_{\boldsymbol{\theta}}(\mathbf{x}_t \mid \mathbf{y}_{t+1:T})\}_{t=1}^{T-1}$. We will refer to these samples as *positive samples*. We can then draw a second set of samples, but discard the observed data, $\tilde{\mathbf{x}}_{1:T}^{1:M} \sim p_{\boldsymbol{\theta}}(\mathbf{x}_{1:T})$. We will refer to these as *negative samples*. The sets of positive and negative examples form the data on which we will train the twist classifier. Generating examples sequentially in this manner is cheap and parallelizable, allowing us to use relatively large values of $M$.

To train the DRE twist, we first pass the RNN backwards over sampled synthetic observed data $\mathbf{y}_{2:T}^m$ to generate the sequence of encodings $\mathbf{e}_{2:T}^m$ (noting that we flip the resulting encodings so they are "forward" in time). To evaluate the probability of a positive classification we concatenate the encoding $\mathbf{e}_{t+1}^m$ with $\mathbf{x}_t^m$ at each timestep, feed the concatenation into the head MLP, and take the output as a positive example Bernoulli logit. To evaluate the probability of a negative classification we take the same sequence of encodings, $\mathbf{e}_{2:T}^m$, concatenate them at each time step with $\tilde{\mathbf{x}}_{1:T-1}^m$, feed the result at each timestep into the head MLP, and take the result as a negative example Bernoulli logit. These outputs are used to compute the cross-entropy loss as written in Algorithm 1 at each timestep, which we average across time and across positive and negative examples to create the final loss. This approach allows us to estimate an entire sequence of ratios, $\{p_{\boldsymbol{\theta}}(\mathbf{x}_t \mid \mathbf{y}_{t+1:T})/p_{\boldsymbol{\theta}}(\mathbf{x}_t)\}_{t=1}^{T-1}$, for positive and negative examples using a single RNN backwards pass.

### C.5 The SIXO Bound Can Become Tight

**Proposition 1. (Reproduced from Section 3.3)** Sharpness of the SIXO bound. *Let* $p(\mathbf{x}_{1:T}, \mathbf{y}_{1:T})$ *be a latent variable model with Markovian structure as defined in Section 2, let* $\mathcal{Q}$ *be the set of*

*possible sequences of proposal distributions indexed by parameters $\boldsymbol{\theta} \in \Theta$, and let $\mathcal{R}$ be the set of possible sequences of positive, integrable twist functions indexed by parameters $\boldsymbol{\psi} \in \Psi$. Assume that $\{p(\mathbf{x}_t \mid \mathbf{x}_{t-1}, \mathbf{y}_{1:T})\}_{t=1}^T \in \mathcal{Q}$ and $\{p(\mathbf{y}_{t+1:T} \mid \mathbf{x}_t)\}_{t=1}^{T-1} \in \mathcal{R}$. Finally, assume $\mathcal{L}_{\mathrm{SIXO}}(\boldsymbol{\theta}, \boldsymbol{\psi}, \mathbf{y}_{1:T})$ has the unique optimizer $\boldsymbol{\theta}^*, \boldsymbol{\psi}^* = \arg\max_{\boldsymbol{\theta} \in \Theta, \boldsymbol{\psi} \in \Psi} \mathcal{L}_{\mathrm{SIXO}}(\boldsymbol{\theta}, \boldsymbol{\psi}, \mathbf{y}_{1:T})$.*

*Then the following holds:*

1. *$q_{\boldsymbol{\theta}^*}(\mathbf{x}_t \mid \mathbf{x}_{1:t-1}, \mathbf{y}_{1:T}) = p(\mathbf{x}_t \mid \mathbf{x}_{1:t-1}, \mathbf{y}_{1:T})$ for $t = 1, \ldots, T$,*

2. *$r_{\boldsymbol{\psi}^*}(\mathbf{y}_{t+1:T}, \mathbf{x}_t) \propto p(\mathbf{y}_{t+1:T} \mid \mathbf{x}_t)$ up to a constant independent of $\mathbf{x}_t$ for $t = 1, \ldots, T-1$,*

3. *$\mathcal{L}_{\mathrm{SIXO}}^K(\boldsymbol{\theta}^*, \boldsymbol{\psi}^*, \mathbf{y}_{1:T}) = \log p(\mathbf{y}_{1:T})$ for any number of particles $K \geq 1$.*

*Proof.* First we reproduce the proof in the main text that $\mathcal{L}_{\mathrm{SIXO}}(\boldsymbol{\theta}, \boldsymbol{\psi}, \mathbf{y}_{1:T}) \leq p(\mathbf{y}_{1:T})$ for any setting of $\boldsymbol{\theta}, \boldsymbol{\psi}$. As previously, fix $r_{\boldsymbol{\psi}}(x_T) = 1$ and let $\widehat{Z}_{\mathrm{SIXO}}(\boldsymbol{\theta}, \boldsymbol{\psi}, \mathbf{y}_{1:T})$ be the normalizing constant estimator returned by SMC run with targets $\{p(\mathbf{x}_{1:t}, \mathbf{y}_{1:t}) r_{\boldsymbol{\psi}}(\mathbf{y}_{t+1:T}, \mathbf{x}_t)\}_{t=1}^T$ and proposals $\{q_{\boldsymbol{\theta}}(\mathbf{x}_t \mid \mathbf{x}_{1:t-1}, \mathbf{y}_{1:T}))\}_{t=1}^T$. Then,

$$\mathcal{L}_{\mathrm{SIXO}}(\boldsymbol{\theta}, \boldsymbol{\psi}, \mathbf{y}_{1:T}) \triangleq \mathbb{E}\left[\log \widehat{Z}_{\mathrm{SIXO}}(\boldsymbol{\theta}, \boldsymbol{\psi}, \mathbf{y}_{1:T})\right] \tag{31}$$

$$\leq \log \mathbb{E}\left[\widehat{Z}_{\mathrm{SIXO}}(\boldsymbol{\theta}, \boldsymbol{\psi}, \mathbf{y}_{1:T})\right] \tag{32}$$

$$= \log p(\mathbf{y}_{1:T}), \tag{33}$$

where (32) holds by Jensen's inequality and the concavity of $\log$, and (33) holds by the unbiasedness of SMC's marginal likelihood estimator.

Because $\mathcal{L}_{\mathrm{SIXO}}(\boldsymbol{\theta}, \boldsymbol{\psi}, \mathbf{y}_{1:T}) \leq p(\mathbf{y}_{1:T})$ and we assume $\mathcal{L}_{\mathrm{SIXO}}(\boldsymbol{\theta}, \boldsymbol{\psi}, \mathbf{y}_{1:T})$ has a unique optimizer, any setting of $\boldsymbol{\theta}$ and $\boldsymbol{\psi}$ that causes the bound to hold with equality must be $\boldsymbol{\theta}^*, \boldsymbol{\psi}^* = \arg\max_{\boldsymbol{\theta}, \boldsymbol{\psi}} \mathcal{L}_{\mathrm{SIXO}}(\boldsymbol{\theta}, \boldsymbol{\psi}, \mathbf{y}_{1:T})$. Thus, we conclude the proof by showing that the twists $\{p(\mathbf{y}_{t+1:T} \mid \mathbf{x}_t)\}_{t=1}^{T-1}$ and proposals $\{p(\mathbf{x}_t \mid \mathbf{x}_{1:t-1}, \mathbf{y}_{1:T})\}_{t=1}^T$ cause the bound to hold with equality.

We proceed by induction on $t$, the timestep in the SMC sweep. We will show that for $t = 1, \ldots, T$, $\widehat{Z}_t = p(\mathbf{y}_{1:T})$. In proving this we will also show that for each $t$, $w_t^k$ either equals 1 or $p(\mathbf{y}_{1:T})$ for $k = 1, \ldots, K$, depending on whether resampling occurred.

For $t = 1$ note that

1. $\gamma_1(\mathbf{x}_1^k) = p(\mathbf{x}_1^k, \mathbf{y}_1) p(\mathbf{y}_{2:T} \mid \mathbf{x}_1^k)$,

2. $\gamma_0 \triangleq 1$,

3. and $q_1(\mathbf{x}_1^k) = p(\mathbf{x}_1^k \mid \mathbf{y}_{1:T})$.

Taken together this implies that the incremental weight $\alpha_1^k$ is

$$\alpha_1^k = \frac{p(\mathbf{x}_1^k, \mathbf{y}_1) p(\mathbf{y}_{2:T} \mid \mathbf{x}_1^k)}{p(\mathbf{x}_1^k \mid \mathbf{y}_{1:T})} = \frac{p(\mathbf{x}_1^k, \mathbf{y}_{1:T})}{p(\mathbf{x}_1^k \mid \mathbf{y}_{1:T})} = p(\mathbf{y}_{1:T}) \tag{34}$$

which does not depend on $k$. Because $w_0^k \triangleq 1$, we have that $w_1^k = w_0^k \alpha_1^k = p(\mathbf{y}_{1:T})$ for all $k$. This in turn implies

$$\widehat{Z_1/Z_0} = \frac{\sum_{k=1}^K w_1^k}{\sum_{k=1}^K w_0^k} = \frac{K p(\mathbf{y}_{1:T})}{K} = p(\mathbf{y}_{1:T}), \tag{35}$$

which when combined with the fact that $\widehat{Z}_0 \triangleq 1$ yields

$$\widehat{Z}_1 = \widehat{Z}_0(\widehat{Z_1/Z_0}) = p(\mathbf{y}_{1:T}). \tag{36}$$

If resampling occurs at the end of step 1, all weights $w_1^{1:K}$ will be set to 1. Thus we have shown that $\widehat{Z}_1 = p(\mathbf{y}_{1:T})$ and $w_1^{1:K} = p(\mathbf{y}_{1:T})$ or 1.

Now assume that $\widehat{Z}_{t-1} = p(\mathbf{y}_{1:T})$ and $w_{t-1}^{1:K}$ equals 1 or $p(\mathbf{y}_{1:T})$. Again, we derive the incremental weights $\alpha_t^k$ by noting that

1. $\gamma_t(\mathbf{x}_{1:t}^k) = p(\mathbf{x}_{1:t}^k, \mathbf{y}_{1:t})p(\mathbf{y}_{t+1:T} \mid \mathbf{x}_t^k)$,

2. $\gamma_{t-1}(\mathbf{x}_{1:t-1}^k) = p(\mathbf{x}_{1:t-1}^k, \mathbf{y}_{1:t-1})p(\mathbf{y}_{t:T} \mid \mathbf{x}_{t-1}^k)$,

3. and $q_t(\mathbf{x}_t^k) = p(\mathbf{x}_t^k \mid \mathbf{x}_{1:t-1}^k, \mathbf{y}_{1:T})$

which yields $\alpha_t^k$ as

$$\alpha_t^k = \frac{p(\mathbf{x}_{1:t}^k, \mathbf{y}_{1:t})p(\mathbf{y}_{t+1:T} \mid \mathbf{x}_t^k)}{p(\mathbf{x}_{1:t-1}^k, \mathbf{y}_{1:t-1})p(\mathbf{y}_{t:T} \mid \mathbf{x}_{t-1}^k)p(\mathbf{x}_t^k \mid \mathbf{x}_{1:t-1}^k, \mathbf{y}_{1:T})} \tag{37}$$

$$= \frac{p(\mathbf{x}_{1:t}^k, \mathbf{y}_{1:T})}{p(\mathbf{x}_{1:t-1}^k, \mathbf{y}_{1:T})p(\mathbf{x}_t^k \mid \mathbf{x}_{1:t-1}^k, \mathbf{y}_{1:T})} \tag{38}$$

$$= \frac{p(\mathbf{x}_{1:t-1}^k, \mathbf{y}_{1:T})p(\mathbf{x}_t^k \mid \mathbf{x}_{1:t-1}, \mathbf{y}_{1:T})}{p(\mathbf{x}_{1:t-1}^k, \mathbf{y}_{1:T})p(\mathbf{x}_t^k \mid \mathbf{x}_{1:t-1}^k, \mathbf{y}_{1:T})} \tag{39}$$

$$= 1 \tag{40}$$

for $k = 1, \dots, K$.

Now there are two cases depending on the value of the weights at the previous timestep. If $w_{t-1}^{1:K} = 1$, then $w_t^k = w_{t-1}^k \alpha_t^k = 1$ for all $k$, implying that $\widehat{Z_t/Z_{t-1}} = 1$. Alternatively, if $w_{t-1}^{1:K} = p(\mathbf{y}_{1:T})$ then $w_t^k = p(\mathbf{y}_{1:T})$ for all $k$ which also implies that $\widehat{Z_t/Z_{t-1}} = 1$. Given that $\widehat{Z_t/Z_{t-1}} = 1$ in both cases, and that $\widehat{Z}_{t-1} = p(\mathbf{y}_{1:T})$, we have that

$$\widehat{Z}_t = \widehat{Z}_{t-1}(\widehat{Z_t/Z_{t-1}}) = p(\mathbf{y}_{1:T}). \tag{41}$$

Finally, if resampling occurs then the weights $w_t^{1:K}$ will be set to 1. Thus we have shown that $\widehat{Z}_t = p(\mathbf{y}_{1:T})$ and $w_t^{1:K} = p(\mathbf{y}_{1:T})$ or 1 for each $t = 1, \dots, T$.

We note that we have used $r_{\boldsymbol{\psi}^*}(\mathbf{y}_{t+1:T}, \mathbf{x}_t) = p(\mathbf{y}_{t+1:T} \mid \mathbf{x}_t)$. In the case where $r_{\boldsymbol{\psi}^*}(\mathbf{y}_{t+1:T}, \mathbf{x}_t) \propto p(\mathbf{y}_{t+1:T} \mid \mathbf{x}_t)$, we have that $\widehat{Z}_t \propto p(\mathbf{y}_{1:T})$ for $t < T$. All the proportionality constants for each $r_{\boldsymbol{\psi}^*}(\mathbf{y}_{t+1:T}, \mathbf{x}_t)$ cancel out in (41), and hence we still obtain $\widehat{Z}_T = p(\mathbf{y}_{1:T})$ as required. $\qquad\square$

Note that we have incidentally shown that all weights are equal at each step of SMC for the optimal proposals and twisting functions. This implies that the variance of the importance weights is minimized (i.e. is 0), and if effective sample size is used to trigger resampling, resampling will never occur.

## D   Experiments

Code for reproduction of all experiments is released here: https://github.com/lindermanlab/sixo.

### D.1   Gaussian Drift Diffusion

#### D.1.1   Model Details

The one-dimensional Gaussian drift-diffusion process has joint distribution:

$$p_{\boldsymbol{\theta}}(\mathbf{x}_{1:T}, \mathbf{y}_{1:T}) = p_{\boldsymbol{\theta}}(\mathbf{x}_{1:T}, y_T) = p_{\boldsymbol{\theta}}(x_1)\left(\prod_{t=2}^{T} p_{\boldsymbol{\theta}}(x_t \mid x_{t-1})\right)p_{\boldsymbol{\theta}}(y_T \mid x_T),$$

$$= \mathcal{N}(x_1 \,;\, \alpha, 1)\left(\prod_{t=2}^{T}\mathcal{N}(x_t \,;\, x_{t-1} + \alpha, 1)\right)\mathcal{N}(y_T \,;\, x_T + \alpha, 1),$$

where the free parameters of the model are $\boldsymbol{\theta} = \{\alpha\} \in \boldsymbol{\Theta} = \mathbb{R}$, the state is $x_t \in \mathcal{X} = \mathbb{R}$, and the observed data are $\mathbf{y}_{1:T} = y_T \in \mathbb{R}$. Training data are sampled from this joint distribution with $\alpha = 1$. Note that the distributions we show in Figure 1 were generated with $\alpha = 0$ for clarity.

### D.1.2 Analytic Forms

The $t^{\text{th}}$ marginal of the filtering distribution for $t < T$ is

$$p_{\boldsymbol{\theta}}(x_t) = \mathcal{N}(x_t \; ; \; t\alpha, t). \tag{42}$$

The $t^{\text{th}}$ marginal of the smoothing distributions can be derived as follows:

$$p(x_t \mid y_T) \propto p(x_t)p(y_T \mid x_t), \tag{43}$$
$$= \mathcal{N}\left(x_t; t\alpha, t\right) \mathcal{N}\left(y_T; x_t + \alpha(T - t + 1), T - t + 1\right),$$
$$= \mathcal{N}\left(x_t; t\alpha, t\right) \mathcal{N}\left(x_t; y_T - \alpha(T - t + 1), T - t + 1\right). \tag{44}$$

Noting that the product of two Gaussian densities is also Gaussian:

$$\mathcal{N}\left(x; \mu_1, \sigma_1^2\right) \mathcal{N}\left(x; \mu_2, \sigma_2^2\right) \propto \mathcal{N}\left(x; \frac{\sigma_2^2 \mu_1 + \sigma_1^2 \mu_2}{\sigma_1^2 + \sigma_2^2}, \frac{\sigma_1^2 \sigma_2^2}{\sigma_1^2 + \sigma_2^2}\right),$$

allows us to combine the two Gaussian distributions in (44):

$$p(x_t \mid y_T) \propto \mathcal{N}\left(x_t; \frac{(T - t + 1)t\alpha + t(y_T - \alpha(T - t + 1))}{t + (T - t + 1)}t, \frac{t(T - t + 1)}{t + (T - t + 1)}\right)$$
$$= \mathcal{N}\left(x_t; \frac{t}{T + 1}y_T, \frac{t(T - t + 1)}{T + 1}\right).$$

Hence the smoothing distribution is a Gaussian distribution with analytically computable mean and variance terms.

The filtering and smoothing distributions are equal at $t = T$. It is interesting to note that for $\alpha = \frac{y_T}{T+1}$, which is the maximum likelihood drift parameter for a *single* datapoint $y_T$, the sequence of filtering and smoothing distributions have the same means. However, the variances are different for all $t < T$; in particular, the smoothing distribution variance peaks in the middle of the timeseries, whereas the variance of the filtering distribution is monotonically increasing for $t < T$, and then drops at $t = T$.

According to Proposition 1, we expect the proposal recovered by SIXO, $q_{\theta_t}$, to match the conditional of the smoothing distribution:

$$q_{\boldsymbol{\theta}_1}(x_1 \mid y_T) = p_{\boldsymbol{\theta}}(x_1 \mid y_T) \qquad = \mathcal{N}\left(x_1 \; ; \; \frac{y_T}{T + 1}, \frac{T}{T + 1}\right) \qquad \text{for } t = 1,$$

$$q_{\boldsymbol{\theta}_t}(x_t \mid x_{t-1}, y_T) = p_{\boldsymbol{\theta}}(x_t \mid x_{t-1}, y_T) = \mathcal{N}\left(x_t \; ; \; \frac{(T - t + 1)x_{t-1} + y_T}{T - t + 2}, \frac{T - t + 1}{T - t + 2}\right) \qquad \text{otherwise}.$$

Note that the mean is an affine function with bias equal to zero.

Furthermore, we also expect the optimal twist distribution to be equal to the true lookahead distribution:

$$r_{\boldsymbol{\psi}_t}(y_T \mid x_t) = p_{\boldsymbol{\theta}}(y_T \mid x_t) = \mathcal{N}\left(y_T \; ; \; x_t + \alpha(T - t + 1), T - t + 1\right) \quad \forall t \in 1, \dots, T - 1. \tag{45}$$

### D.1.3 SIXO Variants for the Gaussian Drift Diffusion

For all experiments we use a Gaussian proposal at each timestep, parameterized as $q_{\boldsymbol{\theta}_t}(x_t \mid x_{t-1}, y_T) = \mathcal{N}(x_t; f_t(x_{t-1}, y_T), \sigma_{qt}^2)$ where $f_t$ is a general affine function of $x_{t-1}$ and $y_T$. There are therefore $4T - 1$ proposal parameters to learn ($T$ biases, $T$ $y_T$ coefficients, $T - 1$ $x_{t-1}$ coefficients, and $T$ variances $\sigma_{qt}^2$).

We test four variants of SIXO:

1. **SIXO-u** learns $\boldsymbol{\theta}$ and $\psi$ by gradient ascent on the unified bound given in (3) using the unbiased gradients (reparameterization and resampling gradients). We parameterize the twists as $r_t(y_T, x_t) = \mathcal{N}(y_T; g_t(x_t), \sigma_{rt}^2)$ for $t < T$, where $g_t$ is a learnable affine function and $\sigma_{rt}^2$ is also learned.

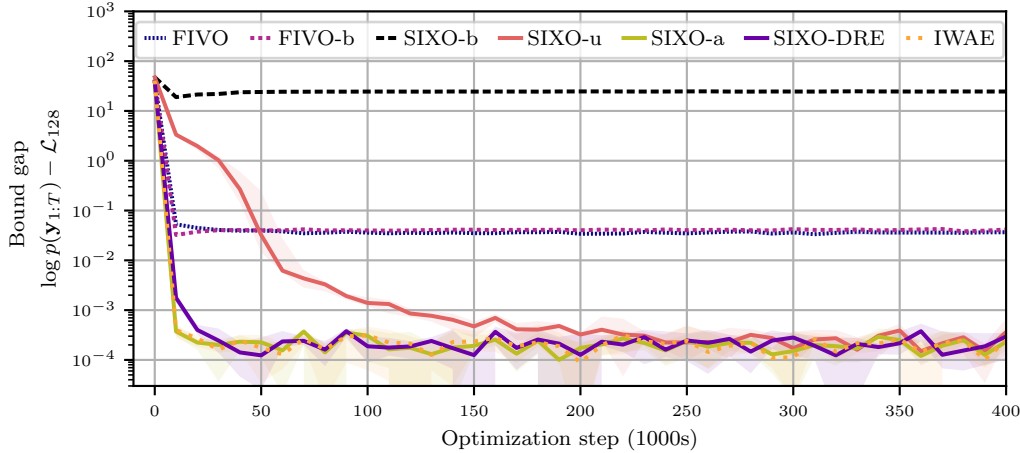

**Figure 6:** Convergence of the bound for all methods discussed in Section D.1. Of these lines, FIVO-b, SIXO-b and SIXO-a were omitted from Figure 2 the main text. Median and quartiles across ten random seeds are shown.

2. **SIXO-DRE** as defined in Algorithm 1 learns $\boldsymbol{\theta}$ by ascending the bound (3) *using the biased reparameterization gradients* as in (18). The twist parameters $\boldsymbol{\psi}$ are then fit using a density ratio update. The twist is parameterized as an MLP that produces the coefficients of a quadratic function over $\mathbf{x}_t$ as a function of $y_T$ and $t$. This quadratic function is evaluated at $\mathbf{x}_t$ to compute the $\log r_{\psi}$ value.

3. **SIXO-a** (not included in Figure 2) uses the analytic form for the twist as a function of $\boldsymbol{\theta}$ and $y_T$ (specified in (45)). There are no free parameters to learn for this twist, and $\boldsymbol{\theta}$ is learned by ascending the bound (3) using the biased reparameterization gradients in (18).

4. **SIXO-b** (not included in Figure 2) learns $\boldsymbol{\theta}$ and $\boldsymbol{\psi}$ by gradient ascent on the unified bound given in (3) using biased reparameterization gradients (18). We parameterize the twists as $r_t(y_T, x_t) = \mathcal{N}(y_T; g_t(x_t), \sigma_{rt}^2)$ for $t < T$, where $g_t$ is a learnable affine function and $\sigma_{rt}^2$ is also learned.

Note that the true distributions lie within the variational families (assuming a sufficiently expressive MLP for SIXO-DRE, which is not unreasonable). In all of these models we initialize the parameter $\alpha = 0$.

### D.1.4 Results

We compare the four different variants of SIXO to IWAE, FIVO with biased gradients, and FIVO with unbiased gradients. For clarity, we omitted some of these comparisons from Figure 2 in the main text, but include them here in Figure 6. Individual seeds for each experiment were run using two CPU cores, 8Gb of memory, and had a runtime of no longer than five hours. We sample 1,000 synthetic trajectories of length $T = 10$ from the model and perform minibatch stochastic gradient descent using the ADAM optimizer [55] with a learning rate of $1 \times 10^{-3}$, a minibatch size of 32, and 10 particles. In the case of SIXO-DRE we alternate between 1,000 updates to the model and proposal and 1,000 twist updates, where the tilt minibatch size is 64. We report and plot the means across ten random seeds. The variances for FIVO, FIVO-b, SIXO-b are too small to be seen.

The model drift $\alpha$ is initialized to zero, affine function weights are initialized to zero, affine function biases are initialized to zero, and $\sigma_{qt}^2$ and $\sigma_{rt}^2$ are initialized to one.

In Figure 6 we show the convergence of the bound across all seven methods we considered. FIVO with unbiased gradients (FIVO) performs comparably to FIVO with biased gradients (FIVO-b), however, converges slightly more slowly. We find that SIXO-b performed worse than all the other methods, similar to the results in Lawson et al. [11]. We therefore omitted it from the main paper for brevity. However, this result motivated us to find alternative methods.

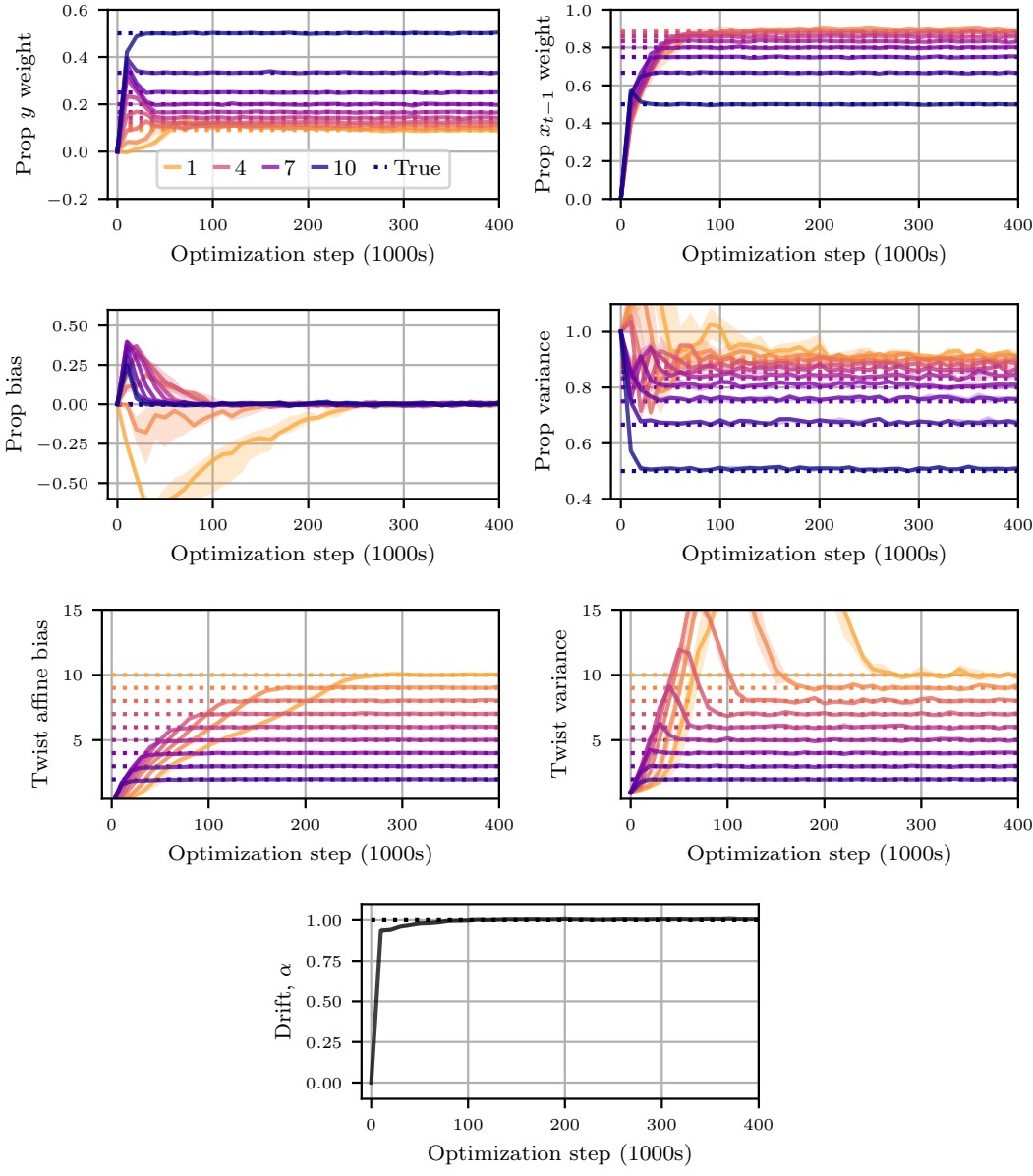

**Figure 7:** Convergence for all free parameters in the GDD experiment using SIXO-u. The drift parameter ($\alpha$) is constant across time. Coloring of lines indicates the time $t \in 1, \ldots, 10$. Median and quartiles across ten random seeds are shown.

All of SIXO-u, SIXO-DRE and SIXO-a converge to the correct solution and achieve a tight variational bound (parameter convergence is shown in Figure 7 for SIXO-u). SIXO-u converges to a tight bound most slowly, but SIXO-DRE and SIXO-a converge quickly at a rate similar to IWAE. This result shows that SIXO is able to recover the optimal model, true posterior, and true twists, and that it can recover a tight variational bound. It also shows that SIXO-DRE converges in a rate commensurate with the best-case convergence of SIXO-a. Future work will investigate whether the use of a twist makes the reparameterization gradient less biased.

Figures 1c and 1d show the lineages for two SMC sweeps: one using a model and proposal learned using FIVO, and one using a model, proposal and twist learned using SIXO-u. We resample at every timestep using systematic resampling. In each color we show the unweighted filtering particles prior to resampling (i.e. the propagated particles) at time $t$, and then the smoothing particles after

performing a backwards pass from that timestep. As such, we are showing the family of smoothing and predictive distributions. We see that FIVO proposes particles towards the observation, but that these particles are often preferentially resampled back towards the prior distribution. This results in gross particle degeneracy. In contrast, the lineages for SIXO are nearly perfect, with every particle being resampled exactly once.

## D.2 Stochastic Volatility Model (SVM)

### D.2.1 Model Details

The SVM models an $N$-dimensional state-space is defined as follows:

$$p_{\boldsymbol{\theta}}\left(\mathbf{x}_{1:T}, \mathbf{y}_{1:T}\right) = p_{\boldsymbol{\theta}}\left(\mathbf{x}_1\right) p_{\boldsymbol{\theta}}\left(\mathbf{y}_1 \mid \mathbf{x}_1\right) \prod_{t=2}^{T} p_{\boldsymbol{\theta}}(\mathbf{x}_t \mid \mathbf{x}_{t-1}) \, p_{\boldsymbol{\theta}}(\mathbf{y}_t \mid \mathbf{x}_t), \tag{46}$$

$$\mathbf{x}_1 \sim \mathcal{N}\left(\mathbf{0}, \mathbf{Q}\right), \quad \mathbf{x}_t = \boldsymbol{\mu} + \boldsymbol{\phi}\left(\mathbf{x}_{t-1} - \boldsymbol{\mu}\right) + \boldsymbol{\nu}_t, \quad \mathbf{y}_t = \boldsymbol{\beta} \exp\left(\frac{\mathbf{x}_t}{2}\right) \boldsymbol{\varepsilon}_t, \tag{47}$$

where the states and observations are defined as:

$$\mathbf{x}_{1:T} = \mathbf{x}_{1:T} \in \mathcal{X}^T = \mathbb{R}^{T \times N}, \quad \mathbf{y}_{1:T} = \mathbf{y}_{1:T} \in \mathcal{Y}^T = \mathbb{R}^{T \times N}, \tag{48}$$

the transition and observation noise terms are defined as:

$$\boldsymbol{\nu}_t \sim \mathcal{N}\left(\mathbf{0}, \mathbf{Q}\right), \quad \boldsymbol{\varepsilon}_t \sim \mathcal{N}\left(\mathbf{0}, \mathbb{I}^{N \times N}\right). \tag{49}$$

and all multiplications are performed element-wise. The model has free parameters defined as:

$$\theta = \left\{\boldsymbol{\mu}, \boldsymbol{\phi}, \boldsymbol{\beta}, \mathbf{Q}\right\}, \quad \text{where} \quad \boldsymbol{\mu} \in \mathbb{R}^N, \quad \boldsymbol{\phi} \in [0,1]^N, \quad \boldsymbol{\beta} \in \mathbb{R}^N_+, \quad \mathbf{Q} \in \mathrm{diag}(\mathbb{R}^N_+), \tag{50}$$

such that there are $4N$ model parameters. The model learning objective is to recover the free parameters, $\boldsymbol{\theta}$, given observed data $\mathbf{y}$. The data we consider are the monthly returns from $N = 22$ currencies over the period from 9/2007 to 8/2017, transformed into the log domain. As a result, $\mathcal{X} = \mathcal{Y} = \mathbb{R}^{119 \times 22}$.

### D.2.2 Results

We use a fixed proposal per timestep parameterized as a Gaussian perturbation to $p_{\boldsymbol{\theta}}$ as in [9],

$$q_{\boldsymbol{\theta}}(\mathbf{x}_t \mid \mathbf{x}_{t-1}, \mathbf{y}_{1:T}) \propto p_{\boldsymbol{\theta}}(\mathbf{x}_t \mid \mathbf{x}_{t-1}) \mathcal{N}(\mathbf{x}_t; \boldsymbol{\mu}_t, \boldsymbol{\Sigma}_t). \tag{51}$$

Thus the parameters specific to $q$ are the means $\boldsymbol{\mu}_t \in \mathbb{R}^N$ and covariance matrices $\boldsymbol{\Sigma}_t \in \mathrm{diag}(\mathbb{R}^N_+)$ for $t = 1, \ldots, T$.

SIXO-q uses a parameterless quadrature twist, so we skip twist optimization and use biased reparameterization gradients of the SIXO bound to fit the model and proposal. For the quadrature twist we use Gauss-Hermite quadrature with degree five.

For SIXO-DRE, we model the twist using the backwards RNN method introduced in Section 3.2. The twist is parameterized with a one-layer LSTM with 128 hidden units and a one-layer MLP with 128 hidden units. The twist is learned using the alternating DRE method described in Section 3.2. We generate a batch of 32,000 synthetic trajectories from the current model, and perform minibatch stochastic gradient descent using the ADAM optimizer [55] with a learning rate of $3 \times 10^{-3}$ and a minibatch size of 64. We apply 1,000 twist updates (corresponding to two epochs) and then apply 1,000 updates to the model and proposal. The model and proposal parameters are updated using the ADAM optimizer [55] with a learning rate of $1 \times 10^{-4}$. We use four particles per SMC sweep, and average across four datasets per model and proposal update.

Individual FIVO and SIXO-q seeds for each experiment were run using four CPU cores, 24Gb of memory, and had a wallclock time of no longer than twenty four hours. SIXO-DRE had a longer runtime of five days, however, competitive results were achieved within two days.

**Train Bound Performance** We compare four methods: IWAE (learning $\boldsymbol{\theta}$), FIVO (learning $\boldsymbol{\theta}$), SIXO-q (learning $\boldsymbol{\theta}$ with a quadrature twist) and SIXO-DRE (learning $\boldsymbol{\theta}$ and $\psi$ using a DRE twist). All methods (other than IWAE) use the biased resampling gradients of (18). We show the median and

quartiles across five random seeds. Each $\mu_n$ is initialized from $\mathcal{N}(0, 0.3)$ (with $n \in 1, \ldots, 22$). $\phi_n$ is learned in the unconstrained space $\mathbb{R}$, and is transformed to $[0, 1]$ by passing the raw $\phi$ through a hyperbolic tangent function. The unconstrained $\phi_n$'s are initialized from $\mathcal{N}(\text{arctanh}(0.1), 0.3)$, and $\log \beta_n$ is initialized from $\mathcal{N}(\log 1.0, 0.3)$. Finally, $\log Q_n$ is initialized from $\mathcal{N}(\log 1.0, 0.3)$. The train bound value we report is taken as the average train bound after $75\%$ of training. We also report $\mathcal{L}_{\text{BPF}}^{2048}$ as the bound evaluated using a BPF with 2,048 particles. This tests the performance of the learned model opposed to inference performance.

A one-way ANOVA [42] rejected the null hypothesis that the mean train bounds are equal ($p < 0.0001$), and a post-hoc Dunnett T3 test [56] found that all methods are statistically significantly different ($p < 0.01$ for all pairs). For the $\mathcal{L}_{\text{BPF}}^{2048}$ values, a one-way ANOVA failed to reject the null hypothesis that the train bounds are equal ($p = 0.26$), so all entries are bolded. This methodology was recommended in Sauder and DeMars [57].

**Test Set Performance**    We also report the performance on a held-out dataset constructed from the new data since Naesseth et al. [9] was published. The test $\mathcal{L}_{\text{BPF}}^{2048}$ we report is the bound evaluated using a BPF with 2,048 particles, averaged across all checkpoints after $75\%$ of training. A one-way ANOVA rejected the null hypothesis that the mean test bounds are equal ($p < 0.0001$), and a post-hoc Dunnett T3 test found the mean SIXO-DRE bound to be the highest ($p < 0.01$ for all pairs).

### D.3    Hodgkin-Huxley Model

We provide a brief overview of the model here for completeness, but refer the reader to Chapter 5.6 of Dayan and Abbott [44] for more detailed information.

#### D.3.1    Model Details

The HH model is a physiologically grounded model of neural action potentials [43] defined through a set of four nonlinear differential equations. Each neuron is defined by four state variables: the instantaneous membrane potential $v(t) \in \mathbb{R}$, the potassium channel activation $n(t) \in [0, 1]$, sodium channel activation $m(t) \in [0, 1]$, and sodium channel inactivation $h(t) \in [0, 1]$. The channel states represent the aggregated probability that the given channel is active. The state evolves according to:

$$C_m \frac{\mathrm{d}v(t)}{\mathrm{d}t} = i_{\text{ext}}(t) - g_L(v(t) - E_L) - g_K n^4(v(t) - E_K) - g_{Na} m^3 h(v(t) - E_{Na}). \quad (52)$$

The membrane capacitance $C_m$ is often defined to be 1.0. The first term, $i_{\text{ext}}(t)$ is the externally injected current. The second term represents the net current through cell membrane due to the potential difference between the intracellular and extracellular mediums, often referred to as the *leakage current*. This current is a function of the membrane capacitance, $g_L$, and the potential of the extracellular medium, $E_L$, where the potential difference across the membrane is then $v(t) - E_L$. The third term represents the net current through the membrane as a result of the potassium channels as a function of the channel state $n(t)$, the channel capacitance $g_K$, and the potassium reversal potential $E_K$. The final term represents the current through the membrane as a result of the sodium channel states, both $m(t)$ and $h(t)$, the sodium channel conductance $g_{Na}$, and the sodium reversal potential $E_{Na}$. The channel states evolve according to:

$$\frac{\mathrm{d}n(t)}{\mathrm{d}t} = \alpha_n(v(t))(1 - n) - \beta_n(v(t))n, \quad (53)$$

$$\frac{\mathrm{d}m(t)}{\mathrm{d}t} = \alpha_m(v(t))(1 - n) - \beta_m(v(t))m, \quad (54)$$

$$\frac{\mathrm{d}h(t)}{\mathrm{d}t} = \alpha_h(v(t))(1 - n) - \beta_h(v(t))h, \quad (55)$$

where $\alpha_n, \alpha_m, \alpha_h, \beta_n, \beta_m, \beta_h$ are all fixed scalar functions of the membrane potential. We discretize this continuous-time differential equation into a discrete-time latent variable model by integrating using Euler integration with an integration timestep of 0.02ms (similarly to Huys and Paninski [47]). This defines the deterministic time-evolution of the neural state.

We follow a similar approach as Huys and Paninski [47] and add Gaussian random noise to each of the four states at each timestep. The membrane potential is unconstrained, and so we can add

noise directly. The gate states, however, are constrained to the range $[0, 1]$. Huys and Paninski [47] use truncated Gaussian noise to avoid pushing the state outside the constrained range. We use a different approach and transform the constrained states into an unconstrained state by applying the inverse sigmoid function to the raw gate value. This has the effect of modifying the variance of the perturbation in constrained space as a function of the state (heteroscedastic noise in constrained space). However, this hetereoscedasticity carries with it a favorable intuition. The magnitude of the noise term is reduced (after being pushed through a sigmoid) close the limits. This means that the same noise kernel provides smaller perturbations close to the extremes, while still retaining a larger permissible perturbations in the mid-range. The model is integrated with a timestep of 0.02ms. We add zero-mean Gaussian noise to the potential with variance scaled by the integration timestep, $\sigma_v^2 = 9\text{mVs}^{-1} \times 0.02\text{ms} = 0.18\mu\text{V}$. The unconstrained gate variables are perturbed by zero-mean Gaussian noise with variance also scaled by the integration timestep, $\sigma_{\{n,m,h\}}^2 = 0.1\text{s}^{-1} \times 0.02\text{ms} = 0.000002$. Observations are sampled from a Gaussian emission distribution centered on the current membrane potential with variance 25mV. Observations are generated every 50 timesteps, corresponding to an acquisition rate of 1kHz.

We initialize the potential according to a Gaussian distribution with mean equal to -65mV and a standard deviation of 25mV. The unconstrained gate variables are initialized from a Gaussian distribution with mean defined by an estimate of the steady-state value $x = \text{sigmoid}\left(\alpha_x(-65)/\alpha_x(-65)-\beta_x(-65)\right)$, where $x$ represents the $n$, $m$ or $h$ states.

To iterate the model, we first constrain the state by passing the gate variables through a sigmoid. The potential is already unconstrained and so requires no transform. We then iterate the model given the constrained state. The iterated state is then unconstrained by passing the gate states through the logit function (inverse of the sigmoid function). The noise term is then added to the iterated, unconstrained state. Observations are then generated by sampling from the emission distribution every 50 steps. We generate traces with 2,048 timesteps, corresponding to approximately 40ms.

### D.3.2 Results

For the experiments presented in Section 5.3 we use either a bootstrap, filtering or smoothing proposal. The bootstrap proposal is defined as $q_{\boldsymbol{\theta}}(\mathbf{x}_t \mid \mathbf{x}_{t-1}, \mathbf{y}_{1:T}) = p_{\boldsymbol{\theta}}(\mathbf{x}_t \mid \mathbf{x}_{t-1})$. The filtering proposal runs an RNN forward over the observed data to create an encoding of all previous observations. This encoding is then concatenated with a Transformer-style embedding of the elapsed time since the last observation [58], and is input into an MLP to produce the residual density with is multiplied with the prior density from which particles are then proposed (a type of *resq* proposal [59]). The smoothing proposal runs two separate RNNs forwards and backwards over the data to encode all previous and future observations. Both RNN encodings and the time encoding are concatenated and used as input to an MLP. For SIXO, the twists are parameterized by an RNN running backwards over the observations. The hidden state of the RNN is concatenated with a time encoding and fed into an MLP to produce the twist value. All RNNs in these experiments have a hidden state of size 32, and all MLPs have two hidden layers of size 32. Individual seeds for each experiment were run using three CPU cores, 9Gb of memory, and had a runtime time of no longer than five days.

**Inference** In Figure 3 we compare the inference performance of a BPF (equivalent to FIVO-bs), to SIXO-bs. The true generative model was used in both cases. This highlights the advantage of using a learned twist under a fixed proposal.

In Table 4 we compare different inference methodologies in the true model across different observation intervals and number of particles. FIVO-fi performs poorly, and drops off dramatically with the number of particles and sampling interval. The performance of SIXO (both bootstrap and smoothing) outperforms FIVO-bs, and is more consistent across the number of particles.

**Model Learning** For the experiments presented in Figure 4 and Table 3 in Section 5.3 we learn the constant external current input, $i_{\text{ext}}$ in (52). We generate 10,000 training sequences, and evaluate on 64 validation and test sequences, all generated with $\theta^{\text{True}} = i_{\text{ext}} = 13\text{mV}$. We use five seeds, deterministically initialized $i_{\text{ext}}$ to a value uniformly spaced between $1.3\mu\text{A}$ and $37.7\mu\text{A}$, corresponding to a relative error of between $-0.9$ and $1.9$. We use $K = 4$ particles per SMC sweep during learning, when evaluating the model and proposal gradients (cf. Line 11 of Algorithm 1), and average across four sequences per parameter update. We use the same proposal and twist

**Table 4:** Comparison of inference performance for different observation intervals and numbers of particles. We normalize $\mathcal{L}^K_{\text{Method}}$ by the number of observations in the sequence for comparison. We compare SIXO-DRE with a smoothing proposal (SIXO-sm), SIXO-DRE with a bootstrap proposal (SIXO-bs), FIVO with a filtering proposal (FIVO-fi), and filtering with a bootstrap proposal (FIVO-bs, equivalent here to a BPF).

| Number of Particles (K) | $\mathcal{L}^K_{\text{Method}}$ / number of observations | | | |
|---|---|---|---|---|
| | SIXO-sm | SIXO-bs | FIVO-fi | FIVO-bs |
| | Observation interval: 2ms | | | |
| 4 | $-5.25 \pm 1.706$ | $-6.77 \pm 1.678$ | $-19.98 \pm 0.206$ | $-15.2 \pm 2.446$ |
| 8 | $-2.90 \pm 0.118$ | $-3.56 \pm 0.539$ | $-20.92 \pm 0.449$ | $-7.91 \pm 0.528$ |
| 16 | $-2.07 \pm 0.097$ | $-2.21 \pm 0.238$ | $-21.57 \pm 0.558$ | $-3.95 \pm 0.520$ |
| 32 | $-1.66 \pm 0.054$ | $-1.75 \pm 0.100$ | $-22.80 \pm 0.457$ | $-2.57 \pm 0.297$ |
| 64 | $-1.50 \pm 0.028$ | $-1.54 \pm 0.050$ | $-22.02 \pm 0.248$ | $-1.69 \pm 0.072$ |
| 128 | $-1.43 \pm 0.013$ | $-1.45 \pm 0.010$ | $-21.31 \pm 0.364$ | $-1.51 \pm 0.049$ |
| 256 | $-1.41 \pm 0.008$ | $-1.41 \pm 0.005$ | $-20.58 \pm 0.341$ | $-1.44 \pm 0.014$ |
| | Observation interval: 1ms | | | |
| 4 | $-2.50 \pm 1.948$ | $-4.56 \pm 1.941$ | $-16.14 \pm 0.843$ | $-16.1 \pm 1.688$ |
| 8 | $-1.30 \pm 0.011$ | $-2.24 \pm 0.561$ | $-14.93 \pm 0.498$ | $-8.28 \pm 0.618$ |
| 16 | $-1.21 \pm 0.017$ | $-1.41 \pm 0.176$ | $-13.29 \pm 0.301$ | $-4.00 \pm 0.612$ |
| 32 | $-1.18 \pm 0.004$ | $-1.36 \pm 0.206$ | $-11.86 \pm 0.304$ | $-2.36 \pm 0.356$ |
| 64 | $-1.17 \pm 0.004$ | $-1.21 \pm 0.045$ | $-10.11 \pm 0.219$ | $-1.43 \pm 0.081$ |
| 128 | $-1.17 \pm 0.002$ | $-1.18 \pm 0.008$ | $-8.58 \pm 0.129$ | $-1.27 \pm 0.048$ |
| 256 | $-1.16 \pm 0.002$ | $-1.17 \pm 0.001$ | $-7.00 \pm 0.192$ | $-1.20 \pm 0.015$ |
| | Observation interval: 0.5ms | | | |
| 4 | $-2.12 \pm 1.182$ | $-4.05 \pm 1.522$ | $-5.73 \pm 0.149$ | $-15.4 \pm 1.997$ |
| 8 | $-1.20 \pm 0.024$ | $-1.83 \pm 0.393$ | $-5.48 \pm 0.107$ | $-7.76 \pm 0.837$ |
| 16 | $-1.11 \pm 0.035$ | $-1.18 \pm 0.096$ | $-5.62 \pm 0.168$ | $-3.88 \pm 0.620$ |
| 32 | $-1.06 \pm 0.013$ | $-1.17 \pm 0.254$ | $-6.09 \pm 0.302$ | $-2.17 \pm 0.305$ |
| 64 | $-1.04 \pm 0.014$ | $-1.05 \pm 0.017$ | $-5.93 \pm 0.236$ | $-1.31 \pm 0.066$ |
| 128 | $-1.03 \pm 0.006$ | $-1.03 \pm 0.005$ | $-5.72 \pm 0.298$ | $-1.10 \pm 0.043$ |
| 256 | $-1.02 \pm 0.002$ | $-1.02 \pm 0.001$ | $-5.19 \pm 0.233$ | $-1.04 \pm 0.015$ |
| | Observation interval: 0.2ms | | | |
| 4 | $-1.28 \pm 0.185$ | $-3.57 \pm 1.369$ | $-2.36 \pm 0.081$ | $-13.71 \pm 1.753$ |
| 8 | $-1.00 \pm 0.009$ | $-1.57 \pm 0.362$ | $-2.43 \pm 0.205$ | $-6.69 \pm 0.800$ |
| 16 | $-0.95 \pm 0.018$ | $-1.02 \pm 0.081$ | $-2.43 \pm 0.075$ | $-3.37 \pm 0.416$ |
| 32 | $-0.92 \pm 0.003$ | $-1.00 \pm 0.209$ | $-2.49 \pm 0.106$ | $-1.93 \pm 0.243$ |
| 64 | $-0.91 \pm 0.010$ | $-0.91 \pm 0.009$ | $-2.31 \pm 0.097$ | $-1.17 \pm 0.082$ |
| 128 | $-0.90 \pm 0.002$ | $-0.90 \pm 0.007$ | $-2.02 \pm 0.059$ | $-0.97 \pm 0.030$ |
| 256 | $-0.90 \pm 0.001$ | $-0.89 \pm 0.001$ | $-1.79 \pm 0.089$ | $-0.92 \pm 0.011$ |

architectures described above. We take 400 steps in the model per 100 steps in the twist. We perform a hyperparameter search over the learning rate for the model, proposal, and twist, and a learning rate decay schedule that halves the learning rate at specific steps. The optimal parameters are shown in Table 5. We selected the best hyperparamters based on validation-set accuracy, and then evaluated the

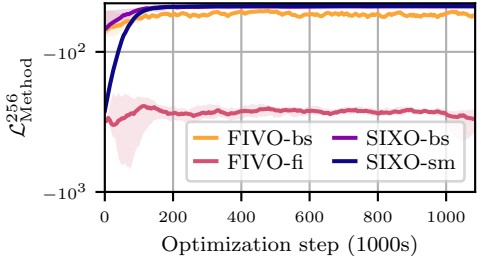

**Figure 8:** Full version of Figure 4a showing the poor performance of FIVO-fi.

final three checkpoints of each selected run on the test set 10 times to produce $\mathcal{L}_{\text{BPF}}^{256}$ and $\mathcal{L}_{\text{Method}}^{256}$ values shown.

**Table 5:** Learning rate hyperparameters for the model learning experiment presented in Section 5.3.

| Method | Model LR | Proposal LR | Twist LR | Learning rate decay schedule (1000s) |
|--------|----------|-------------|----------|--------------------------------------|
| FIVO-bs | 0.01 | N/A | N/A | [150, 300, 600] |
| FIVO-fi | 0.0003 | 0.0002 | N/A | [100, 200, 400] |
| SIXO-bs | 0.01 | N/A | 0.0005 | [800, 1600, 3200] |
| SIXO-sm | 0.0003 | 0.0002 | 0.003 | [80, 160, 320] |