# OpenReview forum: "SIXO: Smoothing Inference with Twisted Objectives"
_NeurIPS.cc/2022/Conference — NeurIPS 2022 Accept_

### Official Review · Reviewer_7TcZ · 2022-07-10

**Rating:** 8
**Confidence:** 4
**Soundness:** 3 good
**Presentation:** 4 excellent
**Contribution:** 3 good

**Summary:**

The paper proposes Smoothing Inference for Twisted Objectives (SIXO) for sequential Monte Carlo inference using approximations of the full posterior as the target distribution at each stage.  Importance weights are computed by learning approximations of the posterior (up to proportionality).  These approximations are learned by using variational backwards methods to solve for twisting functions that approximate the predictive distribution on future measurements, given the current state.  The authors propose learning the twists by casting the ratio of backwards messages and latent marginals as a density ratio estimation problem, leading to SIXO-DRE.  Empirical results are presented on small-scale simulation models and are compared to Filtering Variational Objectives (FIVO).

**Questions:**

The paper argues that target distributions in SMC is a design choice, yet the target distributions of SMC are uniquely defined to be the sequence of marginal or joint filter distributions (i.e. conditional on past observations only).  Under what circumstances are these targets a valid choice in algorithm development?

**Limitations:**

The authors do not clearly discuss limitations of the methodology.  This reviewer sees no obvious negative societal impact of the proposed work.

**Strengths And Weaknesses:**

**STRENGTHS**
The paper is well-written, well-motivated, and empirical results are sufficient for the stated claims.  The content is technically sound and appears to be without significant errors.  Figures are particularly well prepared, easily readable fonts, lines, legends, and colors.  Overall, it is a strong paper.

**WEAKNESSES**
The argument for choosing posteriors as the target—instead of filter distributions—largely relies on a non-gradual shift in the target distributions.  The authors motivate this with the pathological case of a drift diffusion whereby a single observation is obtained at the final timestep.  However, typical dynamical systems receive measurements along with latent states at each time, and thus filters gradually shift.  In such settings as the stochastic volatility model one does not observe a strong benefit over FIVO, for example.  As a result, it is unclear to what extent SIXO is beneficial in typical settings where observations are dense in the time domain.

The consistency results in Proposition 1 are largely vacuous.  Assumptions of the propositions state that the space of twists $\mathcal{R}$ must include the true predictive distribution, and that the space of proposals $\mathcal{Q}$ must include the true posterior (conditioned on the previous state).  These conditions cannot be verified in general and typically hold only in trivial settings.

Full posterior target distributions are impractical in the setting where data arrive in a streaming fashion with inference being updated as new data arrive.  Such a setting is the most common one for SMC and related inference for dynamical systems.  The proposed SIXO requires static data for updating backwards messages, and so cannot be applied in streaming scenarios.  This could not, for example, be used in a control scenario, where particle filters are often applied.  The authors do not discuss this limitation and instead rely on static data for experiments and model parameter learning.

Detailed comments below:
* Line 157 : Typo, change $p_{\theta}(x_t \mid y_{t:1:T})$ to $p_{\theta}(x_t \mid y_{t+1:T})$
* The authors propose SIXO-u then state that gradient estimators are high variance and thus impractical.  Yet there exist numerous variance reduction techniques that can be applied in this setting (indeed, FIVO uses such approaches).  It seems that the authors prematurely abandon this idea.

---

> ### Author Response · Authors · 2022-08-02
> **Review Response 1 of 2**
>
> Thank you for your detailed review. We appreciate the chance to dialogue with you and improve our work. We have responded to your shared concerns in the general response, and address specific issues below.
>
> **It is unclear whether SIXO would be beneficial in situations where observations are dense in time.**
>
> We agree this was unclear to us as well, and was a motivating factor in the design of our experiments, specifically the inclusion of the SVM.
>
> Even with dense observations, the smoothing distributions can provide significant additional information compared to the filtering distributions, making targeting the smoothing distributions preferable over the targeting filtering distributions.
>
> With the SVM, we observe SIXO providing a significant benefit over FIVO, roughly the same size benefit (in nats) that FIVO provides over IWAE.
>
> Furthermore, even if SIXO did not provide benefits in models with dense observations, there are many models of interest without dense observations that SIXO could be used for. Examples include the Hodgkin-Huxley model, ladder VAEs [1], diffusion models [2], and others.
>
> **The claims of proposition 1 are largely vacuous.**
>
> The statement was modeled after a similar proposition in the FIVO paper [3] that motivated us to pursue the SIXO project, specifically that with FIVO the true posterior is not the optimum of the bound except in extremely uncommon model structures. In SIXO, the true posterior and twist *are* optima of the bound, as derived in Proposition 1, and in our opinion it is valuable to know that FIVO’s pathologies are not present in SIXO.
>
> Furthermore, almost all variational algorithms must confront the fact that the true posterior does not lie in the proposed variational family –– SIXO is no different in this regard. SIXO does, however, place very mild additional assumptions on the model, namely that the log-density ratio of the backwards message and dynamics prior must be expressible as a neural network. Note that this does not place explicit distributional assumptions on the model, and in practice the twist family can be made arbitrarily expressive by increasing the size of this neural network.
>
> **SIXO is not useful for streaming settings, and more generally we did not discuss the limitations of SIXO.**
>
> Thank you for noting this, we agree that SIXO-DRE is not useful for streaming settings. We have included a section on SIXO’s limitations in the main text of the paper that discusses this (along with other limitations).
>
> **We did not attempt to apply variance reduction techniques to the unbiased score-function gradient estimators, and perhaps abandoned SIXO-u too soon.**
>
> This was one aspect of the project that was difficult to convey in the finished paper. In reality, there was extensive experimental evaluation of SIXO-u, both by us and by prior work such as TVSMC [4].
>
> Designing variance-reduction techniques for the score function gradients is complicated by the fact that the resampling steps of SMC make the particles depend on each other. Baselines generally work by estimating the “reward” of a trajectory (in this case the sum of future log probabilities of the observations) using only information up to the current timestep. Because the resampling step makes the trajectories depend on each other, a ‘leave one out’ baseline is impossible after the next resampling step. Put another way, constructing a baseline for a single particle out of average rewards of other particles is impossible because the other particles are not statistically independent of the original particle in question. TVSMC tried the single-step leave one out baseline for SIXO-u (called TVSMC-rgrad in their paper), and found it to not work. We tried similar-in-spirit exponential moving average baselines and found their performance poor as well.
>
> Another option is a parametric baseline that attempts to train a function approximator to predict the future reward of a particle given its current state. We tried these, but found them difficult to train. They also added a lot of complexity given that the method needed four function approximators: the model, the proposal, the tilt, and the baseline.
>
> Yet more options include the 'relaxed' gradient estimators such as the Gumbel-softmax/concrete distribution [5,6]. We tried several of these, but found their performance extremely poor. We suspect that the bias in the gradient estimate compounds over the length of the sequence, making the gradient estimates for early timesteps unusable.
>
> In summary, we found that attempting to reduce the variance of the SIXO-u objective resulted in complicated methods with extremely poor performance. Indeed, most attempts did not train at all. We have added a discussion of these attempts in the supplement.
>
> (Continued in a response to this comment.)

---

> > ### Author Response · Authors · 2022-08-02
> > **Review Response 2 of 2**
> >
> > **Are the target distributions considered a design decision in SMC algorithms?**
> >
> > The targets are often considered a design decision for SMC, and are chosen based on the requirements of the algorithm. We agree that often the most sensible choice of targets are either the filtering or smoothing distributions, but there is a rich history of other choices.
> >
> > “A Tutorial on Particle Filtering and Smoothing: Fifteen years later” by Arnaud Doucet and Adam M. Johansen [7] writes “We emphasize that this [filtering] is just one particular choice of target distributions. Not only can SMC methods be used outside the filtering context but, more importantly for this tutorial, some advanced particle filtering and smoothing methods discussed below do not rely on this sequence of target distributions.”
> >
> > Two concrete examples of this are:
> >
> > - Auxiliary particle filtering [8] or fixed-lag smoothing in general can be interpreted as SMC with normalized targets that condition on a fixed window of future observations.
> >
> > - Filtering SMC with MCMC moves [9] runs filtering SMC, but interleaves each SMC timestep with an MCMC step on the particle trajectories. This can greatly reduce particle degeneracy. As stated in the Doucet and Johansen tutorial [7], filtering SMC with MCMC moves can also be interpreted as SMC with target distributions on an extended space that admit the filtering distributions as marginals over the particle trajectories.
> >
> > Thank you again for your thorough review, it was very helpful. If you have any additional questions we are more than happy to discuss further.
> >
> > -The SIXO authors
> >
> > ## References
> >
> > [1] Casper Kaae Sønderby, Tapani Raiko, Lars Maaløe, Søren Kaae Sønderby, and Ole Winther. "Ladder variational autoencoders." Advances in neural information processing systems 29 (2016).
> >
> > [2] Yang Song, Jascha Sohl-Dickstein, Diederik P. Kingma, Abhishek Kumar, Stefano Ermon, and Ben Poole. "Score-based generative modeling through stochastic differential equations." arXiv preprint arXiv:2011.13456 (2020).
> >
> > [3] Chris J. Maddison, Dieterich Lawson, George Tucker, Nicolas Heess, Mohammad Norouzi, Andriy Mnih, Arnaud Doucet, and Yee Teh. "Filtering variational objectives." Advances in Neural Information Processing Systems 30 (2017).
> >
> > [4] Dieterich Lawson, George Tucker, Christian A. Naesseth, Chris Maddison, Ryan P. Adams, and Yee Whye Teh. "Twisted variational sequential monte carlo." In Third workshop on Bayesian Deep Learning (NeurIPS). 2018.
> >
> > [5] Eric Jang, Shixiang Gu, and Ben Poole. "Categorical reparameterization with gumbel-softmax." arXiv preprint arXiv:1611.01144 (2016).
> >
> > [6] Chris J. Maddison, Andriy Mnih, and Yee Whye Teh. "The concrete distribution: A continuous relaxation of discrete random variables." arXiv preprint arXiv:1611.00712 (2016).
> >
> > [7] Arnaud Doucet and Adam M. Johansen. "A tutorial on particle filtering and smoothing: Fifteen years later." Handbook of nonlinear filtering 12, no. 656-704 (2009): 3.
> >
> > [8] Michael K. Pitt and Neil Shephard. "Filtering via simulation: Auxiliary particle filters." Journal of the American statistical association 94, no. 446 (1999): 590-599.
> >
> > [9] Walter R. Gilks and Carlo Berzuini. "Following a moving target—Monte Carlo inference for dynamic Bayesian models." Journal of the Royal Statistical Society: Series B (Statistical Methodology) 63, no. 1 (2001): 127-146.

---

### Official Review · Reviewer_L368 · 2022-07-11

**Rating:** 8
**Confidence:** 4
**Soundness:** 3 good
**Presentation:** 3 good
**Contribution:** 4 excellent

**Summary:**

The paper introduces a new form of sequential Monte Carlo for particle smoothing problems. The basic idea is to learn the twisting functions that convert the filtering distributions to approximate smoothing distributions using the density estimation ratio trick. This provides a potentially tight bound on the marginal likelihood.


**Questions:**

Does the approach scale to high dimensional settings such as, for example, natural video processing?


**Limitations:**

The paper would benefit from a longer discussion section, discussing potential limitations and failure cases. I cannot see any negative societal impacts.


**Strengths And Weaknesses:**

Strengths:
- The basic idea is solid and simple
- The method has appealing theoretical properties. The tightness of the bound is a clear benefit over competing SMC approaches.
- The empirical evaluation covers several interesting timeseries problems and compares performance with relevant baselines. The method clearly outperforms the baselines.
- The paper is clear and well-written

Weaknesses:
- It would have been interesting to see the method applied to high dimensional problems

---

> ### Author Response · Authors · 2022-08-02
> **Review Response**
>
> Thank you for your review of our paper and thoughtful comments! Please see our general response to all reviewers where we addressed some of your shared concerns. If you have any outstanding questions, we are happy to discuss them further. We address your specific questions below.
>
> **Does SIXO scale to high-dimensional data?**
>
> We agree, scaling to higher dimensional examples is an interesting topic. Please note that the SVM example has 22 dimensional latent states; while this is not as high dimensional as some deep generative models for sequential data, it poses a nontrivial inference and learning problem. We are actively working on experiments using other deep generative models and datasets. We hope to have results ready before the end of the rebuttal period, and will update our response either way.
>
> **The paper should include a discussion of the limitations of SIXO.**
>
> We agree, and as explained in the general response have included a thorough discussion of SIXO’s limitations in the main text of the paper.
>
> Thank you again for taking the time to review our submission.
>
> -The SIXO authors.

---

### Official Review · Reviewer_4LSp · 2022-07-12

**Rating:** 8
**Confidence:** 3
**Soundness:** 4 excellent
**Presentation:** 4 excellent
**Contribution:** 4 excellent

**Summary:**

Users of sequential Monte Carlo algorithms have many choices to make, including the proposal distributions and the sequence of intermediate target distributions. An appealing approach is to attempt to learn optimal proposals and intermediate targets by
stochastic gradient descent; this can be formulated within a variational inference framework, where the SMC algorithm can be understood
either as *being* the variational family, or as a technique for tightening an existing variational bound. Although researchers have had some success applying this approach to the problem of learning proposals, learning intermediate targets is more challenging: they affect the objective's value only through their impact on SMC's resampling steps, which are ignored by commonly used biased gradient estimators.

This paper presents a new method, SIXO, for learning the intermediate target distributions in sequential Monte Carlo (or rather, learning twisting functions that tilt filtering distributions toward the true smoothing distributions, which are the optimal intermediate targets in a state-space model). The key innovation is to recognize that the optimal twisting function in this setting is a density ratio, and to apply density ratio estimation techniques. In particular, the authors propose to train a classifier to distinguish between samples from $p(x_{t}) p(y_{t+1:T})$ and $p(x_{t}) p(y_{t+1:T} \mid x_t)$. The logits from the trained classifier can be understood as computing, up to a constant, the log density ratio needed for twisting. The authors further propose parameterizing the classifier with a recurrent architecture, reading the sequence of $y$'s backward from the end.

In experiments, the authors show that SIXO outperforms methods that do not use twisting, and that their density-ratio estimation scheme outperforms an unbiased gradient estimator for the joint optimization of proposal, model, and twist function parameters.

**Questions:**

1. In Figure 3, why is resampling so much more frequent in the SIXO plot than in the BPF plot? From the trajectories I would guess that ESS is relatively high throughout the SIXO sweep, and sometimes dips quite low in the BPF sweep, which should lead to the opposite behavior (more resampling in BPF).

2. The related work draws connections to APF, iAPF, cSMC, etc. and explains that SIXO is novel with respect to them (e.g., it supports model learning). But if I am not trying to learn model parameters, and am just trying to get better intermediate targets, how should I understand the relative benefits and disadvantages of these methods, compared to SIXO?

**Limitations:**

Yes, I think the paper adequately addresses the limitations. (However, see Weakness 2 -- I think additional context could help readers make better sense of the experimental results.)

**Strengths And Weaknesses:**

Strengths:

1. This is a very nice paper that proposes an appealing solution to a long-standing and important problem. Although framed in the context of model-learning and variational inference, the technique seems like it would also be useful for improving SMC inference even when the model is not learned.

2. The idea to treat the twisting function as a density ratio estimator gives rise to a simple and efficient training procedure.

3. Parameterizing twisting functions as recurrent networks is a smart choice that greatly reduces computational complexity.

4. The paper is very nicely written with well-made and informative figures. I especially liked Figure 2, which clearly shows the problem with FIVO: the good proposals it learns are rendered useless by resampling against inaccurate intermediate targets.


Weaknesses:

1. Because the twisting functions are trained on samples of (x, y) from the generative model, they may work less well when the true observations look quite different from the synthetic data. (This may be less of an issue in the setting where the model is learned to fit the true observations, however.)

2. I don't have a great sense of how hard the inference problems from the experiments are; it might have been nice to report results from the IWAE baseline on all of them, to help readers understand to what extent resampling is even necessary on these problems.


Not weaknesses, but minor suggestions:

* The paper could do more to highlight the challenge that the DRE solution overcomes. The move from FIVO / VSMC to what the paper calls "SIXO-u" is quite small; FIVO/VSMC jointly optimize any parameters of the SMC algorithm, and so of course we could introduce learned parameters into the intermediate targets. The challenge is that learned targets affect the SMC algorithm only through the resampling steps, which are not differentiable; as a result, all signal for improving the intermediate targets' parameters comes from high-variance score function estimator terms, which previous papers have advocated dropping during training. This explains why there isn't a "SIXO-biased" that just optimizes the SIXO objective directly with a biased estimator -- the biased estimator does not enable training of the intermediate targets.

* On L145-148 a reference to the appendix where the architecture is more fully explained might be nice. (A possible misunderstanding readers might have is to think X is an input to the recurrent part of the network, but of course it is important that it not be--you run the recurrent part once to get the embeddings, then score many X's using those embeddings during SMC.)

* Possible connection for related work -- Nested Variational Inference (https://proceedings.neurips.cc/paper/2021/hash/ab49b208848abe14418090d95df0d590-Abstract.html) also attempts to learn intermediate targets in SMC-like algorithms, and also tries to do it using a separate objective for training them that is not understood as part of one overall variational bound.

---

> ### Author Response · Authors · 2022-08-02
> **Review Response Part 1 of 2**
>
> Thank you again for taking the time to review our paper and providing valuable feedback. We have addressed your points below.
>
> **SIXO could be used for inference, not just model learning.**
>
> We agree, using SIXO for inference is a great topic for future work. This work started by extending FIVO so we felt model learning was an important aspect, but we are actively working on evaluating SIXO for inference as future work.
>
> **Training the twist on samples from the model could lead to poor performance when the training data and model samples look very different.**
>
> We agree, and were worried about that initially as well. To combat this, we designed importance-weighted objectives that leveraged training data instead of model samples to train the twists. However, it turned out that we did not need this additional complexity to train good twists for the models and data we considered in the paper. We suspect that extensions along these lines may be important for applying SIXO in other domains, however.
>
> **How hard are these inference problems, and how does IWAE perform on them?**
>
> Please see the general response for these results.
>
> **You could discuss the challenges DRE overcomes, specifically the gradient bias issues.**
>
> Thanks for this suggestion, we agree that better situating SIXO and DRE in the current literature around twist estimation is important. As this was something commented on by multiple reviewers, we have added a section to the main text that discusses this. One small difference is that we think there is some signal to the twist even without resampling gradients because the twist values appear in the weights and thus the bound SIXO is ascending. We believe the quality of this signal is prohibitively low, making twist training without some form of resampling gradients impossible. This is supported both by our own experiments and Figure 1a of the TVSMC paper [2], which shows the twist values changing over training even without resampling gradients.
>
> **A more detailed explanation of the architecture would be helpful.**
>
> We agree! We have added more detail to the appendix and referenced it in the main text. We specifically highlight the fact that the backwards RNN is run only once per sequence.
>
> **In Figure 3 (the Hodgkin-Huxley model) why does SIXO resample more frequently than FIVO?**
>
> This is a great observation with an interesting explanation. The particle weights in a bootstrap particle filter (BPF) don’t change between observations (which in our Hodgkin-Huxley simulation occur once every 50 steps). Thus, resampling events can only be triggered at observations for the BPF. In contrast, with SIXO the twist can change particle weights between observations, triggering resampling events even on steps with no observation because the new particles poorly explain _future_ observations. We have added an extra sentence highlighting this to the figure caption.
>
> (Continued in a reply to this comment)

---

> > ### Author Response · Authors · 2022-08-02
> > **Review Response Part 2 of 2**
> >
> > **How does SIXO compare to the iterated auxiliary particle filter (iAPF) and controlled sequential Monte Carlo (CSMC) for inference alone (no model learning)?**
> >
> > We hope to thoroughly explore this question in future work, but can provide some preliminary thoughts.
> >
> > iAPF and CSMC learn using a Bellman equation derived by writing the backwards twist recursively in terms of the same twist one step in the future, see e.g. section 3.3 of the iAPF paper [3] or section 2.2 of the TVSMC paper [2]. The parameters are then updated by descending the mean squared difference between the right- and left-hand sides of the recursion.
> >
> > There are two potential issues with this approach that SIXO avoids. First, computing this recursion involves integrating the model likelihood $p(y_t | x_t)$ against the dynamics $p(x_t | x_{t-1})$. To our knowledge, the iAPF, TVSMC, and CSMC papers [3,2,4] all make the assumption that the log dynamics $\log p(x_t | x_{t-1})$ and log lookahead densities $\log p(y_{t+1:T} | x_t)$ are quadratic in $x_t$. This makes the integral computable analytically, but places severe constraints on the model family. For example, this assumes that the dynamics $p(x_t | x_{t-1})$ are Gaussian. SIXO does not place such restrictions on the model or twist families.
> >
> > Secondly, while the Bellman equation can sometimes be effective, the twist regression targets for each timestep depend on future twists. Specifically, at each timestep the twist is regressed against an expression that integrates the twist at the next timestep against the model dynamics. If the twist at the next timestep is poor, then the targets for the twist at the current timestep will be poor as well. Stated differently, it is impossible to learn early twists before learning the later twists well. SIXO does not suffer from this pathology because all twist timesteps are trained on independent losses that come from a separate density-ratio estimation task.
> >
> > **SIXO seems related to Nested Variational Inference**
> >
> > Thank you for this reference! We agree, it is very related and have added it to the related works.
> >
> > Thank you again for taking the time to review our submission. If there are any outstanding questions, we are happy to discuss them further.
> >
> > -The SIXO authors
> >
> > ## References
> >
> > [1] Christian A. Naesseth, Scott Linderman, Rajesh Ranganath, and David Blei. Variational sequential Monte Carlo. In International Conference on Artificial Intelligence and Statistics, pages 968–977. PMLR, 2018.
> >
> > [2] Dieterich Lawson, George Tucker, Christian A. Naesseth, Chris Maddison, Ryan P. Adams, and Yee Whye Teh. Twisted variational sequential Monte Carlo. In Third workshop on Bayesian Deep Learning (NeurIPS), 2018.
> >
> > [3] Pieralberto Guarniero, Adam M. Johansen, and Anthony Lee. The iterated auxiliary particle filter. Journal of the American Statistical Association, 112(520):1636–1647, 2017.
> >
> > [4] Jeremy Heng, Adrian N. Bishop, George Deligiannidis, and Arnaud Doucet. Controlled sequential Monte Carlo. The Annals of Statistics, 48(5):2904–2929, Oct 2020. ISSN 0090-5364. doi: 10.1214/19-aos1914.

---

> > > ### Comment · Reviewer_4LSp · 2022-08-09
> > > **Thank you!**
> > >
> > > Thank you for the thorough response! Your answers are very clear, and I appreciate the new experiments, which demonstrate that SIXO was tested on inference tasks that do require some form of resampling (justifying the learning of twists). I continue to believe this is a very nice paper and will be maintaining my score at 8.
> > >
> > > ---
> > >
> > > > One small difference is that we think there is some signal to the twist even without resampling gradients because the twist values appear in the weights and thus the bound SIXO is ascending.
> > >
> > > Thanks for this correction—I see now that the story is a bit more complicated. To make sure I understand: if there were no resampling, the twist distributions would not have any effect on the overall weight, because the twist term in the numerator of each weight would be canceled by the term in the denominator of the following increment. However, with resampling, the weights of all particles are averaged after each resampling step, and this averaging prevents the terms from exactly canceling. In the case where resampling is adaptive, only the intermediate targets that immediately precede a resampling step affect the overall weight / receive signal during backprop. And even though they do receive signal, it may not be in the right direction, because the effect of the resampling itself is ignored.

---

> > > > ### Author Response · Authors · 2022-08-09
> > > > **Yes**
> > > >
> > > > Yes, that is our understanding. Thank you again for your review.

---

### Author Response · Authors · 2022-08-02
**General response to all reviewers**

We thank the reviewers for taking the time to review our submission and for providing very helpful feedback.  We presented SIXO, a method for smoothing inference in SMC that learns backwards “twists’’ via density ratio estimation to warp filtering target distributions into the smoothing distributions. We are very pleased that all reviewers understood our work and posed insightful questions.

In this general response we will address concerns raised by multiple reviewers. First, we have fixed multiple typos and generally improved the clarity and narrative of the paper thanks to your suggestions. Second, we have run more experiments that compare SIXO and IWAE. Finally, we have added a discussion of the limitations of our work.

### New Experimental Results

As requested by reviewer 4LSp, we have included IWAE results for both the SVM and HH models. IWAE gradients are unbiased and it leverages smoothing information in its proposal, so it should do quite well on some learning problems; however, the key difference is that IWAE doesn’t have a resampling step, so we expect it to do poorly on high dimensional inference problems. Indeed that is what we see.

The SVM results are (only the IWAE row is new):

| Method   | Train $\mathcal{L}^4_\mathrm{Method}$ | Train BPF 2048 | Test BPF 2048  |
| -------- | -------------- | -------------- | -------------- |
| IWAE     | 6940.29 ± 1.13 | 58.98 ± 0.026  | 3351.21 ± 2.44 |
| FIVO     | 6923.66 ± 5.71 | 58.98 ± 0.0039 | 3352.51 ± 0.14 |
| SIXO-q   | 6930.56 ± 4.76 | 58.98 ± 0.0052 | 3353.10 ± 0.81 |
| SIXO-DRE | 6933.77 ± 3.45 | 58.98 ± 0.0040 | 3354.08 ± 0.67 |

For this model, IWAE obtains the highest train-set $\mathcal{L}^4_\mathrm{Method}$, but it performs the same as all other methods on the training set marginal likelihood estimate and worst on the test set marginal likelihood estimate.

_Note: This IWAE $\mathcal{L}^4_\mathrm{Method}$ number is much higher than the one reported in the VSMC paper [1], which was 6911.2. We suspect the difference is that we trained IWAE for 100x more steps than the VSMC authors. We are able to reproduce their 6911.2 number at the number of steps the VSMC authors trained for. We are actively investigating this and will update with any new information before the rebuttal period ends._

For the Hodgkin-Huxley model, IWAE performs similar in model learning to SIXO, but far underperforms SIXO and FIVO in inference.

| Method   | Test $\mathcal{L}^{256}_\mathrm{Method}$ | Test BPF 256   | Relative Parameter Error  |
| -------- | --------------  | -------------- | -------------- |
| IWAE     | -282.86 ± 11.37 | -48.50 ± 0.33  | 0.02 ± 0.06    |
| FIVO     | -50.25 ± 1.38   | -50.21 ± 1.28  | 0.46 ± 0.16    |
| SIXO     | -47.94 ± 0.25   | -48.53 ± 0.36  | 0.02 ± 0.09    |

We feel that these results do not significantly change the story of the paper, especially as IWAE is known not to scale well relative to FIVO in terms of sequence length or latent dimension.

### SIXO Limitations

A main limitation of SIXO, as noted by reviewer 7TcZ, is that it is offline-only, meaning that it cannot be used in streaming contexts where data for new timesteps arrives continuously. There are straightforward extensions of SIXO to fixed-lag streaming contexts (e.g. using windowed tilts), but we do not consider them in this paper.

Another limitation is that the quality of the twists strongly depends on the difficulty of estimating the density ratio between the backwards messages and dynamics priors. Density ratio estimation can encounter difficulties when the densities in question have near-disjoint support, making the ratio unstable. While truly disjoint support is uncommon in most models, it is often true that the dynamics priors are far broader than the backwards messages as they do not condition on any observations. This can lead to poor density ratio estimates, and thus poor twists. In practice this seems to not be a big issue as we still observe SIXO outperforming FIVO. It is possible that this is more of a problem in certain models not considered here, however.

A further limitation of the approach (which is also somewhat unresolved from the original FIVO/VSMC work) is the effect that the number of particles used has on the bias in the gradient estimator. This effect is present in FIVO/VSMC as well, but was not commented on by the original authors. For this work we limited ourselves to 4 particles for simplicity, but characterization of the precise role of the number of particles is an important topic of future work.

Thank you again for your work in reviewing our paper. We found your comments helpful and insightful.

-The SIXO authors.

---

### Meta-Review · Area_Chair_BMBo · 2022-08-26

**Recommendation:** Accept
**Confidence:** Certain

**Metareview:**

This paper proposes a new method to adjust the proposal used by sequential Monte Carlo. Existing methods struggle for this problem since the resampling step poses difficulties for reparameterization-based methods. This paper proposes a "twisting" method to learn a density ratio by training a classifier to distinguish between two distributions. Reviewers agreed the method was novel, relevant, and sufficiently supported by experiments.

**Award:**

No

---

### Decision · Program_Chairs · 2022-09-14

Accept